# Integration of FRET and sequencing to engineer kinase biosensors from mammalian cell libraries

Longwei Liu[1,9], Praopim Limsakul[1,2,9], Xianhui Meng[3,9], Yan Huang[4], Reed E. S. Harrison [1], Tse-Shun Huang[1,8], Yiwen Shi[1], Yiyan Yu[1], Krit Charupanit [5], Sheng Zhong[1], Shaoying Lu[1], Jin Zhang[6], Shu Chien[1,7], Jie Sun [3✉] & Yingxiao Wang [1✉]

The limited sensitivity of Förster Resonance Energy Transfer (FRET) biosensors hinders their broader applications. Here, we develop an approach integrating high-throughput FRET sorting and next-generation sequencing (FRET-Seq) to identify sensitive biosensors with varying substrate sequences from large-scale libraries directly in mammalian cells, utilizing the design of self-activating FRET (saFRET) biosensor. The resulting biosensors of Fyn and ZAP70 kinases exhibit enhanced performance and enable the dynamic imaging of T-cell activation mediated by T cell receptor (TCR) or chimeric antigen receptor (CAR), revealing a highly organized ZAP70 subcellular activity pattern upon TCR but not CAR engagement. The ZAP70 biosensor elucidates the role of immunoreceptor tyrosine-based activation motif (ITAM) in affecting ZAP70 activation to regulate CAR functions. A saFRET biosensor-based high-throughput drug screening (saFRET-HTDS) assay further enables the identification of an FDA-approved cancer drug, Sunitinib, that can be repurposed to inhibit ZAP70 activity and autoimmune-disease-related T-cell activation.

[1] Department of Bioengineering, Institute of Engineering in Medicine, University of California, San Diego, CA, USA. [2] Center of Excellence for Trace Analysis and Biosensor, Division of Physical Science, Faculty of Science, Prince of Songkla University, Songkhla, Thailand. [3] Department of Cell Biology and Bone Marrow Transplantation Center of the First Affiliated Hospital, Zhejiang University School of Medicine, Zhejiang, P.R. China. [4] Department of Chemistry and Chemical Engineering, Hunan University, Changsha, P.R. China. [5] Department of Biomedical Sciences and Biomedical Engineering, Faculty of Medicine, Prince of Songkla University, Songkhla, Thailand. [6] Department of Pharmacology, University of California, San Diego, CA, USA. [7] Department of Medicine, University of California, San Diego, CA, USA. [8] Present address: BioLegend, San Diego, CA, USA. [9] These authors contributed equally: Longwei Liu, Praopim Limsakul, Xianhui Meng. ✉email: sunj4@zju.edu.cn; yiw015@eng.ucsd.edu

Genetically encoded biosensors based on Förster Resonance Energy Transfer (FRET) have revolutionized the imaging of molecular signals (e.g., protein–protein interactions, protein activations, ion, and small molecule dynamics) in live cells with high spatiotemporal resolution[1,2]. However, the limited sensitivity of these biosensors prevents broader applications in cellular studies and drug screening[3,4]. At present, optimization of FRET biosensors is rather empirical and labor-intensive, and the design of these biosensors is often limited by the availability of accurate protein structures[5]. To address these problems, several methods have been proposed that utilize evolutionary strategies in bacteria and yeasts[3,6–9]. While these methods are very well-designed, they generally need additional selection steps to identify the optimized FRET biosensors since results from purified proteins or bacteria/yeasts cannot be translated directly to biosensor responses in mammalian cells. Directed evolution platforms in mammalian cells have been established to evolve transcription factors and G protein-coupled receptors (GPCRs)[10] and to optimize the brightness and membrane localization of the voltage reporters utilizing an elegant robotic cell picking system integrated with microscopy[11]. Semi-rational design of relatively small-scale libraries (≤100 variants) of FRET biosensors in mammalian cells has also been developed to improve RhoA FRET biosensors[12]. However, no method to our knowledge exists that can engineer and screen relatively large-scale libraries (e.g., tens of thousands or larger) of FRET biosensors directly in mammalian cells for the identification of sensitive biosensors in a high throughput fashion.

FRET biosensors have provided a powerful platform to quantify the dynamics of biochemical[13–15] and biomechanical[16] signaling in T cells. T Cell-based immunotherapy, e.g., CAR-T therapy, has revolutionized cancer treatment. The second-generation designs of chimeric antigen receptors (CARs), containing a CD28 or 4-1BB-derived costimulatory domain at the cytoplasmic tail of CAR have been widely applied in the clinic[17]. Different designs of these CAR molecules, with varying immunoreceptor tyrosine-based activation motifs (ITAMs) at the CAR cytoplasmic tail, have been shown to result in different anti-tumor potencies in vivo[18]. However, the mechanism of ITAM in regulating CAR-T cell functions, which is crucial for the design of next-generation CARs, remains unclear. Tyrosine kinases, including Fyn and ZAP70 kinases, serve as key mediators of ITAM and the TCR/CAR cytoplasmic tail[19,20]. Monitoring these kinases provides a powerful means to study ITAM functions. At the same time, ZAP70 plays critical role in T cell signaling and is involved in a variety of diseases[21–24]. For instance, elevated TCR signaling caused by hypermorphic R360P mutation in ZAP70, leads to clinical autoimmune phenotypes characterized by bullous pemphigoid, proteinuria, and colitis[4]. However, due to the low sensitivity of the currently available ZAP70 FRET biosensors[14,15,25] and the relatively weak intrinsic activities of ZAP70 kinase, which led to difficulties in detection[26], methods are not readily available to measure physiologically relevant ZAP70 activity in live cells to study the role of ITAM in regulating TCR/CAR functions.

Despite the crucial roles of ZAP70 kinase in T-cell functions and related diseases, there is no efficient inhibitor targeting ZAP70. High-throughput drug screening (HTDS), an effective way to identify kinase inhibitors, has been limited mainly to conventional in vitro enzymatic assays[27,28]. FRET biosensors can serve as powerful tools for evaluating kinase inhibitors and their related therapeutic drugs in living cells[29,30]. However, despite the successful integration of FRET biosensors in screening assays for monitoring insulin-receptor activation[31], and PKA kinase activity[32], FRET-based biosensors have not been broadly applied to screen kinase inhibitors, mainly due to the relatively small dynamic ranges of FRET biosensors below the minimum 20%

dynamic range needed for HTDS assays[33]. Furthermore, these conventional FRET-HTDS assays cannot differentiate inhibitors of upstream signaling molecules from those directly targeting the kinase itself. The heterogeneous levels of kinase activities in individual host cells may impose additional noise and difficulty to the FRET-based screening platforms[34]. Hence, a different FRET-screening design with high-sensitivity biosensors is needed to screen kinase inhibitors in a high-throughput manner.

Here, utilizing a self-activating FRET (saFRET) biosensor fused to an active kinase domain, we have developed a method to couple FRET signals to next-generation sequencing (NGS) (FRET-Seq) of biosensor libraries in mammalian cells to improve biosensor performance (Fig. 1a). This FRET-Seq platform has been applied to improve both Fyn and ZAP70 biosensors in a high throughput fashion, with the improved ZAP70 biosensor being further applied to single T cell and CAR-T cell imaging. In addition, we developed a saFRET based-HTDS assay to screen compound libraries and identified efficient ZAP70 kinase inhibitors that can suppress T cell activation engendered from pathological ZAP70 mutations.

## Results

**Engineering of Fyn saFRET biosensor**. A typical FRET-based kinase activity biosensor contains a substrate peptide paired with a phosphoamino acid-binding domain as the sensing unit, which can modulate the FRET efficiency of a pair of fluorescent proteins (FPs) as the reporting unit[2]. In order to develop a system for the screening of biosensor libraries in mammalian cells, we designed saFRET biosensor to minimize the impact of noise introduced by heterogeneously expressed kinases in individual host cells. In this design, an active kinase domain is fused to the FRET biosensor via an EV linker to allow self-activation to dominate the FRET signals (Fig. 1b). For Fyn saFRET biosensor, a Fyn kinase domain was linked to the C-terminus of a Fyn activity reporter that contains a Fyn-kinase-specific peptide substrate (EKIEGTYGVV, from p34cdc2) and SH2 as the phosphoamino acid-binding domain[35]. Replacement of the kinase domain by its kinase-dead version altered the FRET ratio and abolished the Fyn inhibitor PP1-induced dynamic changes of the saFRET biosensor during live-cell imaging (Fig. 1c–e), as well as biosensor phosphorylation (Fig. 1f). These results indicate that the FRET change of the saFRET biosensor is specifically mediated by the active kinase domain. The saFRET biosensor was also reversible as washing out PP1 during imaging led to the recovery of the FRET ratio to the original value, in contrast to the control no-wash group (Supplementary Fig. 1). With the modular design of its sensing unit, this functional Fyn-saFRET biosensor was utilized as a template for the library generation (Fig. 1g).

**FRET-seq strategy for screening biosensor library in mammalian cells**. Given the importance of the amino acids surrounding the target tyrosine residue of the substrate in being recognized by the corresponding kinase[36] and the SH2 domain[37,38], we created two biosensor libraries by randomizing these neighboring residues of the substrate domain, including Library 1 (Lib1: −1, −2, −3, Y) and Library 2 (Lib2: Y,+1, +2, +3) using degenerate primers (NNK)[39], with each library consisting of 32,768 DNA variants (corresponding to 8000 amino acid sequences, Supplementary Fig. 2a). To reduce false-positive selection, we also generated control libraries with dead kinase (K299M mutation[40]) for counter screening. After the generation of a mammalian cell library by infection with viral libraries, individual cells expressing biosensor variants were sorted into low and high FRET ratios (ECFP/FRET ratios) by the fluorescence-activated cell sorter (FACS) (Supplementary Fig. 2b–d). These FACS screening and sorting should enrich cells containing the desired

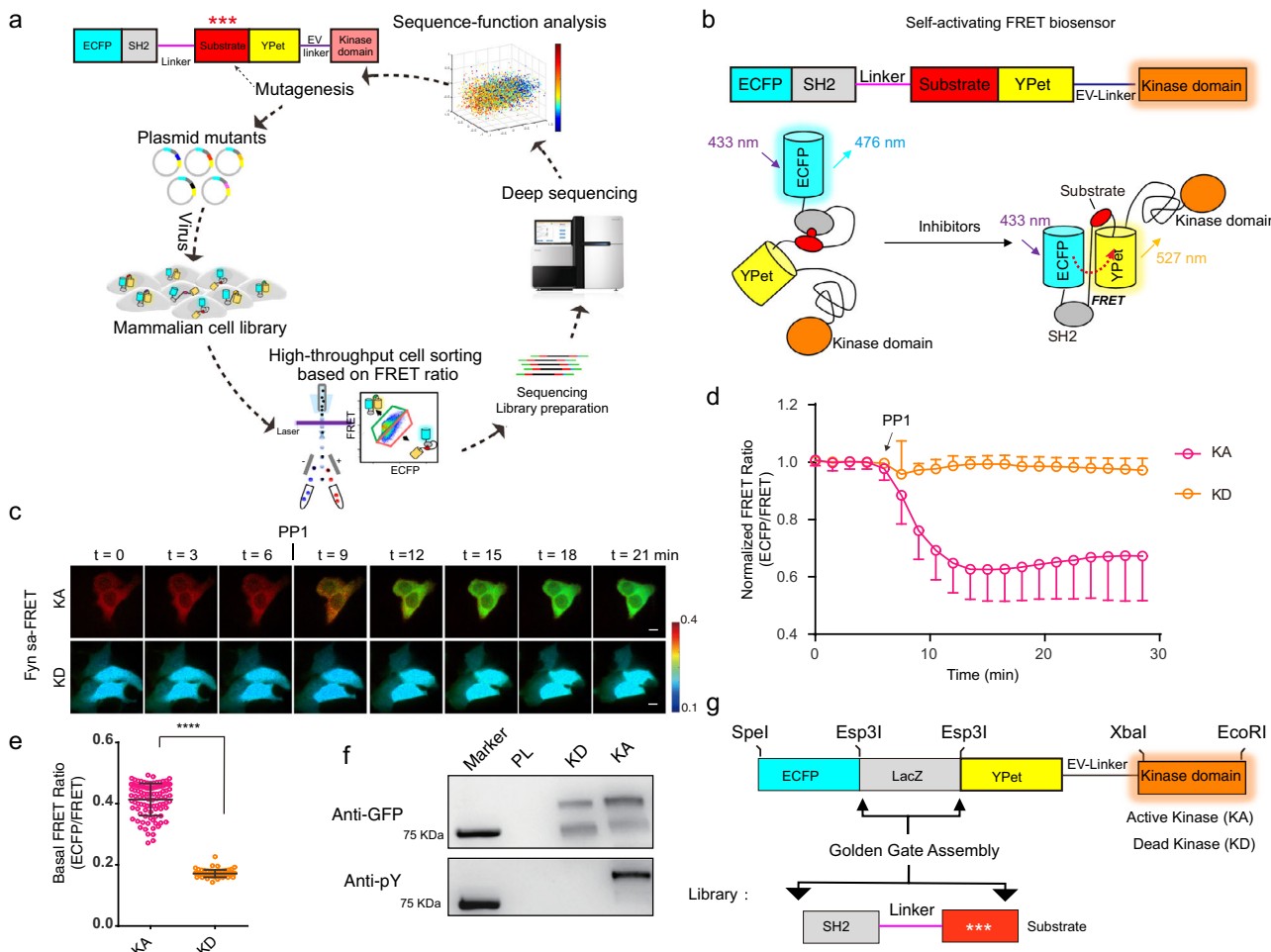

**Fig. 1 Construction and validation of saFRET biosensors. a** Schematics of mammalian cell biosensor library development, screening, and sequencing in mammalian cells. **b** Domain structure and activation mechanism of a saFRET biosensor with a fused kinase domain. **c–d** Representative images (**c**) and time courses (**d**) of Fyn-saFRET biosensor with active kinase domain (KA) (n = 98) or kinase-dead domain (KD) (n = 69) before and after PP1 treatment. Error bars, mean ± SD. Scale bars, 10 μm. The color bar indicates ECFP/FRET emission ratio, with hot and cold colors representing the high and low ratios, respectively. **e** Quantification of the basal FRET ratio of Fyn-saFRET biosensor with KA (n = 98) or KD domain (n = 69). (Unpaired two-tailed Student's t-test, ****P < 0.0001). Error bars, mean ± SD. **f** Western blot analysis of the biosensor phosphorylations. Each biological replicate had similar results (n = 3). **g** Modularized template for the library generation of biosensor variants. The bottom panel illustrates the PCR product of substrate variants using the NNK primer. Source data are provided as a Source Data file.

biosensor variants. The change in frequency of each variant sequence between the FACS-sorted groups and their input control before sorting can represent the enrichment of the variant by sorting[41,42], which can be quantified by calculating the enrichment ratio ($E_v$) of each variant (See supplementary method for details: Sequencing analysis). The ECFP/FRET ratio of a desired saFRET biosensor variant depends on its phosphorylation by kinase; hence it should be (1) high with an active kinase domain (KAH), but (2) low with a kinase-dead domain (KDL)[43]. These desired biosensors should not be enriched in two other libraries, viz. (3) low ratios with an active kinase (KAL) and (4) high ratios with a kinase-dead domain (KDH) (Supplementary Data 1). Using this multiplex sorting strategy in combination with NGS and analysis, we acquired an average of 16 million reads per library and converted the raw reads into amino acid sequences (Fig. 2a). We then applied the four criteria to select biosensor sequences, as illustrated in the four-dimensional plot (Fig. 2b). This approach allowed us to assess the library variants to identify sensitive biosensors with the substrate sequences that allow (a) favorable phosphorylation by the corresponding kinase and (b) efficient binding to the intramolecular SH2 domain upon phosphorylation.

**Identifying the sequences of desired biosensor variants**. Previous reports suggest that the amino acid residues located downstream of tyrosine may be more important in determining the substrate response to kinases and the binding of phosphorylated peptides toward SH2 domains[37,38]. Hence, we first examined the Lib2 variants targeting residues downstream of the consensus tyrosine to verify the selection strategy. Among the forty variants of Fyn biosensors enriched in the KAH group, multiple biosensor variants were identified to have significantly improved dynamic changes comparing to the parent Fyn biosensor, shown by changes of emission ratios upon treatment with PP1, an inhibitor of Src family kinases including Fyn (Fig. 2c–e, Supplementary Fig. 3a–c and Supplementary Data 2). The probability of identifying a better biosensor than the parental biosensor increased from one-dimensional selection (KAH only) to four-dimensional selection after further filtering the variants via KAL, KDH, KDL enrichment criteria (Supplementary Data 2), thus verifying the importance of the multiplex selection strategy. The product of $E_v$ (KAH) and $E_v$ (KDL) was also found to be an efficient ranking factor for the desired biosensor candidates. The parental wild-type (WT) biosensor can be further used as a

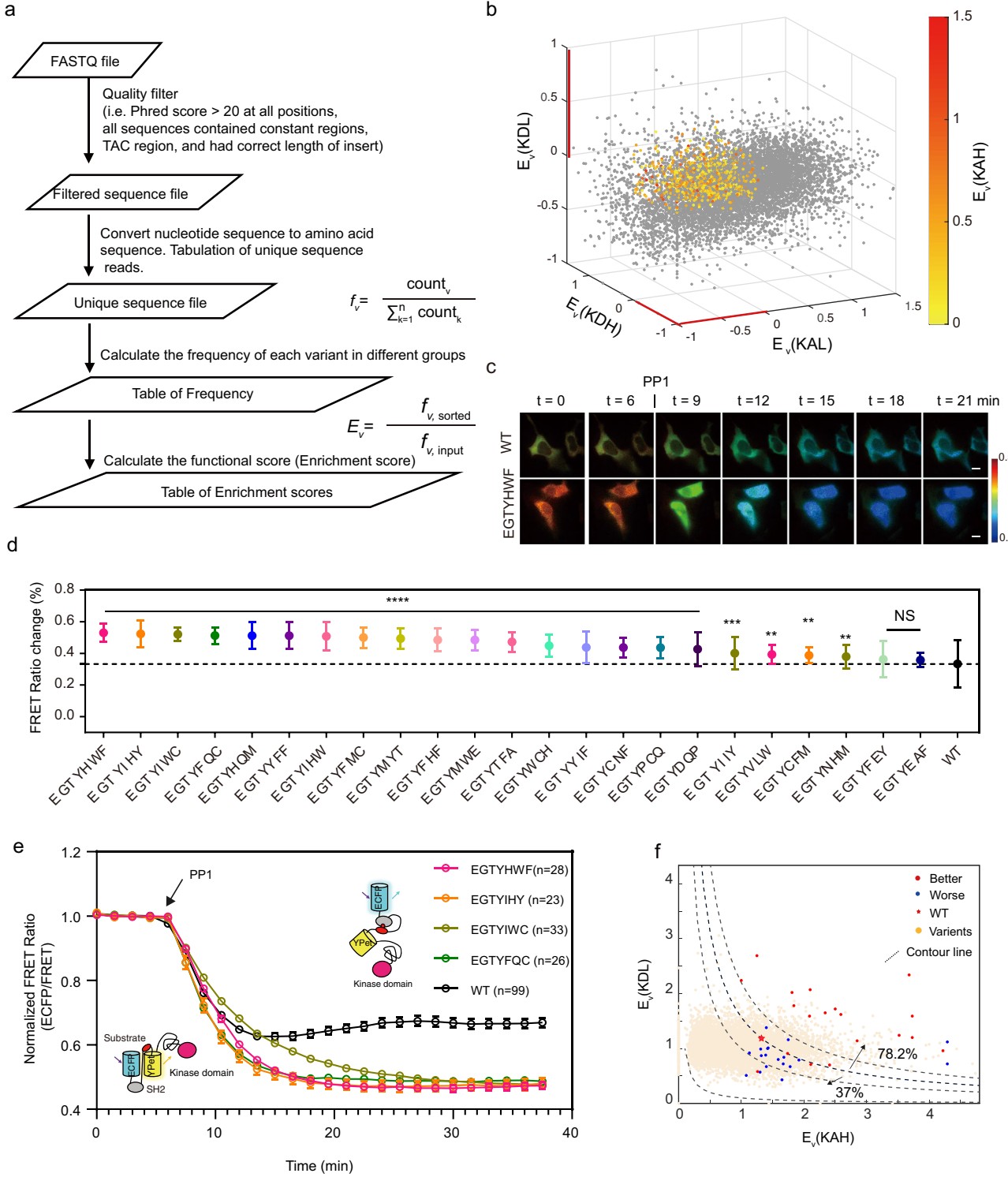

reference to calibrate the biosensors. In fact, the success rate of identifying a biosensor better than the parental WT biosensor increased from 60 to 78% when the product of $E_v$ values of the WT biosensor was used as the threshold in evaluating the 40 tested clones (Fig. 2f and Supplementary Fig. 3d). A selected biosensor with the EKIEGTYHWF substrate sequence demonstrated a ~60% increase in PP1 induced emission ratio changes in HEK cells, compared to the parental biosensor (Fig. 2c–e). With the same filtering and selection approach, the success rate for Lib1 to identify better candidates than the parental biosensor was

significantly lower than that of Lib2 (Supplementary Fig. 4 and Supplementary Data 3 and 4). The combination of two improved mutants from Lib1 and Lib2 did not further improve the performance of the biosensor (Supplementary Fig. 5), potentially due to the uncooperative effect of amino acids up- and down-stream of the consensus tyrosine in the substrate contributing to the recognition by kinase and SH2 domain. Nevertheless, our findings suggest that the FRET-Seq platform combining FRET-based FACS screening with sorting and NGS can directly optimize the Fyn FRET biosensor in mammalian cells.

**Fig. 2 Identification of Biosensors by NGS and sequence-function analysis. a** Workflow of sequencing data analysis. **b** Four-dimensional (4D) plot of the enrichment ratios ($E_v$) of substrate sequences from different sorting groups. The enrichment ratios in KAH group ($E_v$(KAH)) are color-coded, whereas $E_v$(KAL), $E_v$(KDH), and $E_v$(KDL) are plotted along with the three-dimensional coordinates. The selected substrate sequences are highlighted with colors represented by the values of their $E_v$(KAH). **c** Representative time-lapse images of the parental (WT) and improved biosensor (EKIEGTYHWF) before and after PP1 treatment. Scale bars, 10 μm. The color bar indicates ECFP/FRET ratio, with hot and cold colors representing the high and low ratios, respectively. **d** The quantified dynamic changes of biosensor variants (EKIEGTYXXX) upon PP1 treatment (From left to right, $n$ = 28, 23, 33, 26, 33, 33, 30, 28, 22, 29, 28, 32, 23, 19, 21, 25, 32, 19, 19, 25, 35, 30, 18, and 99, respectively, One-way ANOVA, compared to the WT group, ****$P$ < 0.0001, **$P$ = 0.0028, 0.0087, 0.0072, and 0.0093 respectively, NS, not significant with a P = 0.1174 and 0.3005, respectively.). Error bars, mean ± SD. **e** Time courses of normalized ECFP/FRET ratio of the biosensor variants, with that of the parental biosensor labeled in black ($n$ is shown in the figure). Error bars, Mean ± SEM. **f** Mapping of verified substrates in the scatter plot of the enrichment ratios. The dynamic ranges of biosensors were found to have a positive correlation with the product of $E_v$(KAH) and $E_v$(KDL). The percentage indicates the success rate of identifying better biosensors in different product groups (above or below the contour line of the product of WT biosensor). Red and blue dots represent biosensor variants with better and worse performance than the parent biosensor, respectively. The red dotted lines represent the contour lines of the product of $E_v$(KAH) and $E_v$(KDL). Source data are provided as a Source Data file.

**Extending the FRET-seq platform for ZAP70 biosensor optimization.** To extend FRET-seq as a platform to optimize different substrate sequences and kinase FRET biosensors in mammalian cells, we further applied this technology to improve the ZAP70 FRET biosensor, which had low dynamic range that limited its application[14,15,25]. A ZAP70 saFRET biosensor was constructed by fusing ZAP70 kinase domain to a ZAP70 FRET biosensor through the EV linker (Fig. 3a), with the substrate sequences derived from the ZAP70 substrate molecule VAV2[25] or LAT[14,15]. The combination of ZAP70 kinase domain (327–619) and a substrate from LATY191 (SREYVNVSGEL)[24] showed an efficient phosphorylation level of the saFRET biosensor (Supplementary Fig. 6a). The high performance of this combination was further verified by live-cell imaging, in which the saFRET biosensor specifically responded to TAK-659 (25 μM), a moderate inhibitor of ZAP70 kinase[44] (Fig. 3b, c, Supplementary Fig. 6b), but not to PP2, a Src family kinase inhibitor[45] (Supplementary Fig. 6c, d). The FRET change of ZAP70 saFRET biosensor was dominated by the active kinase domain, as evidenced by the observations that the TAK-659-induced dynamic changes (Fig. 3b, c) and phosphorylation (Supplementary Fig. 6a) were abolished when the kinase domain was replaced by its kinase-dead version (K369A)[46]. Hence, we selected the substrate LATY191 and the kinase domain 327-619 for generating the template of ZAP70 saFRET biosensor to develop substrate mutant libraries, including Library 1 (Lib1: −1, −2, −3, Y) and Library 2 (Lib2: Y, +1, +2, +3) (Supplementary Fig. 7).

Using the established FRET-seq platform, the ZAP70 biosensor candidates selected via the four-dimensional plot were further ranked by the product of $E_v$ (KAH) and $E_v$ (KDL) (Fig. 3d–e and Supplementary Data 5). The screening of Lib1 did not lead to a better biosensor (Supplementary Fig. 8 and Supplementary Data 6), consistent with the finding of the Fyn biosensor library. In contrast, six of the selected variants from Lib2 showed significantly higher FRET ratio changes than the parental biosensor upon TAK-659 treatment (Fig. 3f, g). The dynamic range of FRET ratio of the biosensors containing SREYACISGEL or SREYYDMSGEL increased ~50% comparing to the parental one when transiently expressed in HEK cells (Fig. 3g, h). As such, our FRET-seq platform can serve as a high-throughput approach to engineer and develop sensitive kinase FRET biosensors from mammalian cell libraries via the screening of randomized substrate sequences.

**Visualizing T cell signaling with the improved biosensors.** To examine the functionality of our improved biosensors, we removed the kinase domain and applied the ZAP70 biosensor to monitor T cell activation. Our ZAP70 biosensors demonstrated large FRET ratio changes in Jurkat T cells, but not in their ZAP70-deficient derivative, P116 cells, when stimulated with CD3/CD28-antibody clusters to activate the T cell receptor (TCR)[47] (Fig. 4a–e and Supplementary Fig. 9a–c). The selected variant SREYACISGEL exhibited more than fourfold increase in dynamic range at 25 ± 10%, comparing to the parental variant (SREYVNVSGEL) at 5.8 ± 9% (Fig. 4b–e). These results suggest that our biosensor has improved sensitivity and retained specificity in reporting ZAP70 kinase activity.

We further applied our improved biosensor (SREYACISGEL) to visualize the ZAP70 activities in membrane compartments. During T cell activation, TCR becomes phosphorylated by Lck kinase to recruit and activate ZAP70 for the formation of the detergent-resistant microdomains located in lipid rafts[48–50] (Fig. 4f). Our Lyn- and Kras-tagged ZAP70 biosensors were engineered to anchor at the cell membrane, with their Lyn- and Kras-tags targeting rafts and non-raft compartments[51], respectively (Supplementary Fig. 9d). The FRET ratios of the biosensors with Lyn-tag, but not Kras-tag, were significantly elevated upon TCR activation in Jurkat (Fig. 4g, h) or human primary CD4[+] T cells (Supplementary Fig. 9e–f). This observation suggests that the activation of the ZAP70 kinase upon TCR stimulation mainly occurs in membrane rafts, consistent with prior studies[49,50]. The improved biosensor also allowed dynamic imaging of ZAP70 activity during CAR-T/tumor cell engagement (Fig. 4i). Spatially polarized ZAP70 activity was observed to be transiently enriched in the immunological synapse after engagement, but gradually declined during cell detachment and synapse dissolution (Fig. 4j). High-resolution images further revealed that the CAR-T cell synapse was different from that of TCR, as CAR stimulation led to a relatively spread distribution of CAR clusters and ZAP70 activation patterns. In contrast, TCR engagement caused ZAP70 activation with well-organized spatial distribution, concentrating at the central TCR cluster in the immunological synapse region and at the cell periphery where cortical actin fibers accumulate (Supplementary Fig. 9g–h). Hence, our improved biosensors are applicable to visualize the physiological ZAP70 activities in T cells with high spatiotemporal resolutions in both TCR and CAR signaling.

**ITAM regulates CAR functions via ZAP70.** We further applied the optimized ZAP70 biosensor to study how different CAR designs influence CAR-T cell functions. Since both ZAP70 and ERK kinases can be regulated by the CAR cytoplasmic tail and serve as key effectors for CAR signaling and T cell activation[19,20], we examined the role of ITAM motif in regulating the ZAP70 and ERK kinases in response to CAR activation, utilizing our optimized ZAP70 biosensor and an ERK biosensor with high sensitivity[2]. These FRET biosensors were co-expressed with the wild-type CAR (1928ζ, WT-CAR) or its mutated version (1928ζ, XX3-CAR), which had inferior anti-tumor efficacy than its wild-type counterpart (the tyrosine sites of the first two ITAM motifs in the CAR cytoplasmic tail mutated to phenylalanines)[18]. The

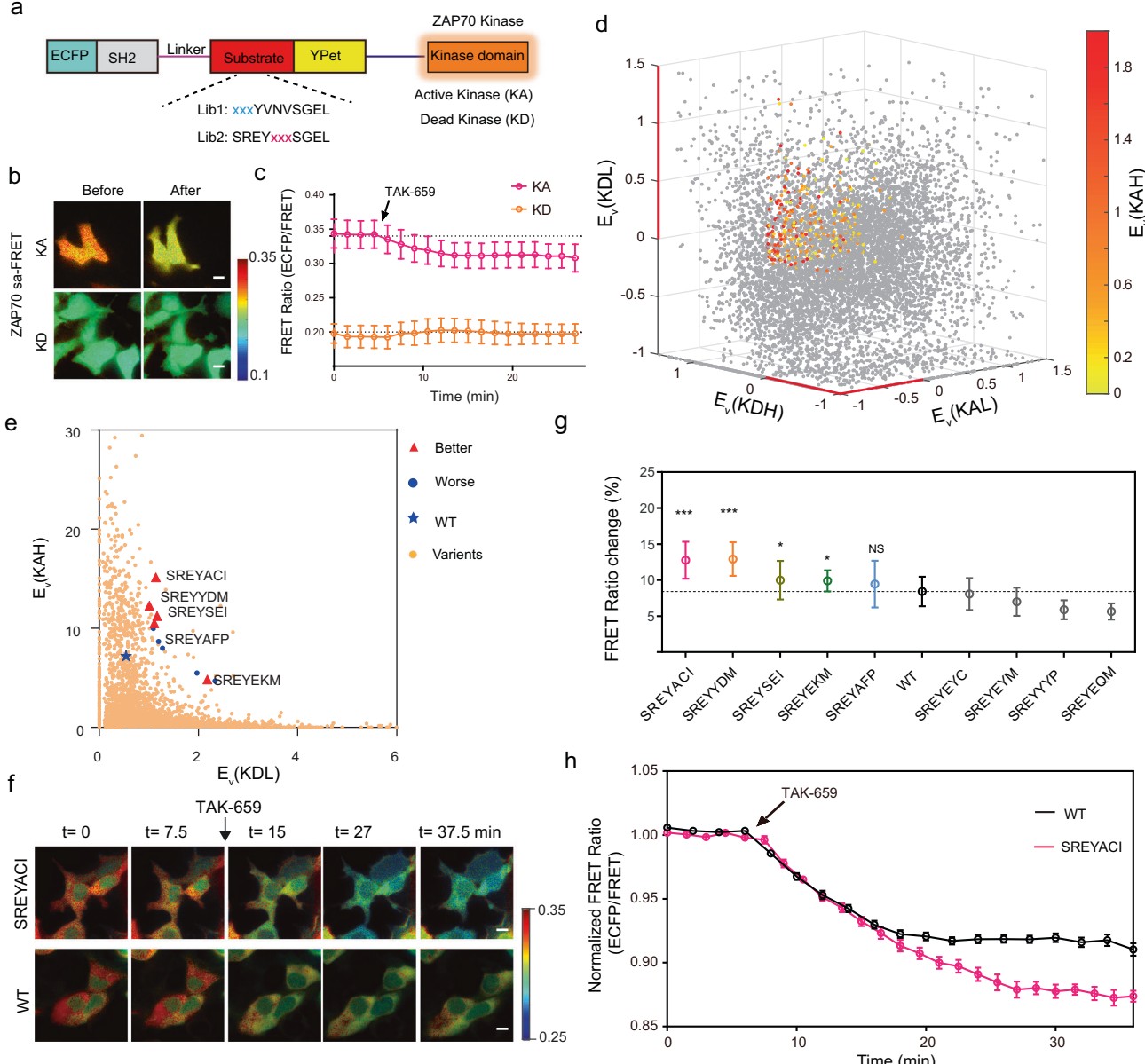

**Fig. 3 Development and optimization of the ZAP70 FRET biosensor. a** Design of the self-activating ZAP70 FRET biosensor as the screening template. **b–c** Representative images (**b**) and time courses (**c**) of FRET ratios of the ZAP70 saFRET biosensor with Active- (KA, $n = 31$) or Dead- (KD, $n = 19$) kinase domain, before and after TAK-659 treatment. Error bars, mean ± SD. Scale bars, 10 μm. See also supplementary video 3. d, The 4D plot of the four enrichment ratios ($E_v$) of substrate sequences. The enrichment ratios in KAH group ($E_v$(KAH)) were color-coded, whereas $E_v$(KAL), $E_v$(KDH), and $E_v$(KDL) are plotted along the three-dimensional coordinates. The selected substrate sequences are highlighted with colors represented by the values of their $E_v$(KAH). **e** Scatter plot of the substrates. The ZAP70 saFRET biosensors with the top 10 highest products of $E_v$(KAH) and $E_v$(KDL) were labeled in red (better biosensors) or blue (worse biosensors). **f** Time-lapse images of the parental (WT) and improved saFRET biosensors after TAK-659 treatment corresponding to the quantified time courses in (**h**). Scale bars, 10 μm. See also supplementary video 4. **g** Percentage changes of saFRET biosensor variants after TAK-659 treatment (From left to right, $n = 26,17,17,15,15,31,18,15,20$, and 15, respectively, One-way ANOVA, compared to the WT group, ***$P < 0.001$, *$P = 0.0192$ and 0.0329, respectively, NS, not significant with a $P = 0.138$, only the varients with a mean value larger than the WT were subjected to statistical analysis). Error bars, mean ± SD. **h** Time courses of FRET ratio of the selected saFRET biosensor variant (SREYACI, n = 26), with that of the parental biosensor (WT, $n = 44$) marked in black. Error bars, Mean ± SEM. **b, f** The color bar indicates ECFP/FRET ratio, with hot and cold colors representing the high and low ratios, respectively. Source data are provided as a Source Data file.

kinase activity was tracked by live-cell imaging after the CAR-T cells expressing either WT-CAR or XX3-CAR were stimulated with antigen-presenting CD19$^+$ 3T3 cells (Fig. 5a). A large FRET ratio change of ERK and ZAP70 biosensors in both types of CAR-T cells was observed when engaging with the CD19$^+$ 3T3 cells; this indicates that ZAP70 and ERK kinases are both responsive to

antigen stimulation (Fig. 5b–g). The ERK signals increased within 5 min after antigen engagement and quickly reached a plateau (Fig. 5b–c). However, no significant difference in ERK activity was observed between CAR-T cells expressing WT-CAR or XX3-CAR (Fig. 5d). In contrast, a significantly delayed response and a reduced activation level of ZAP70 kinase activity were observed in

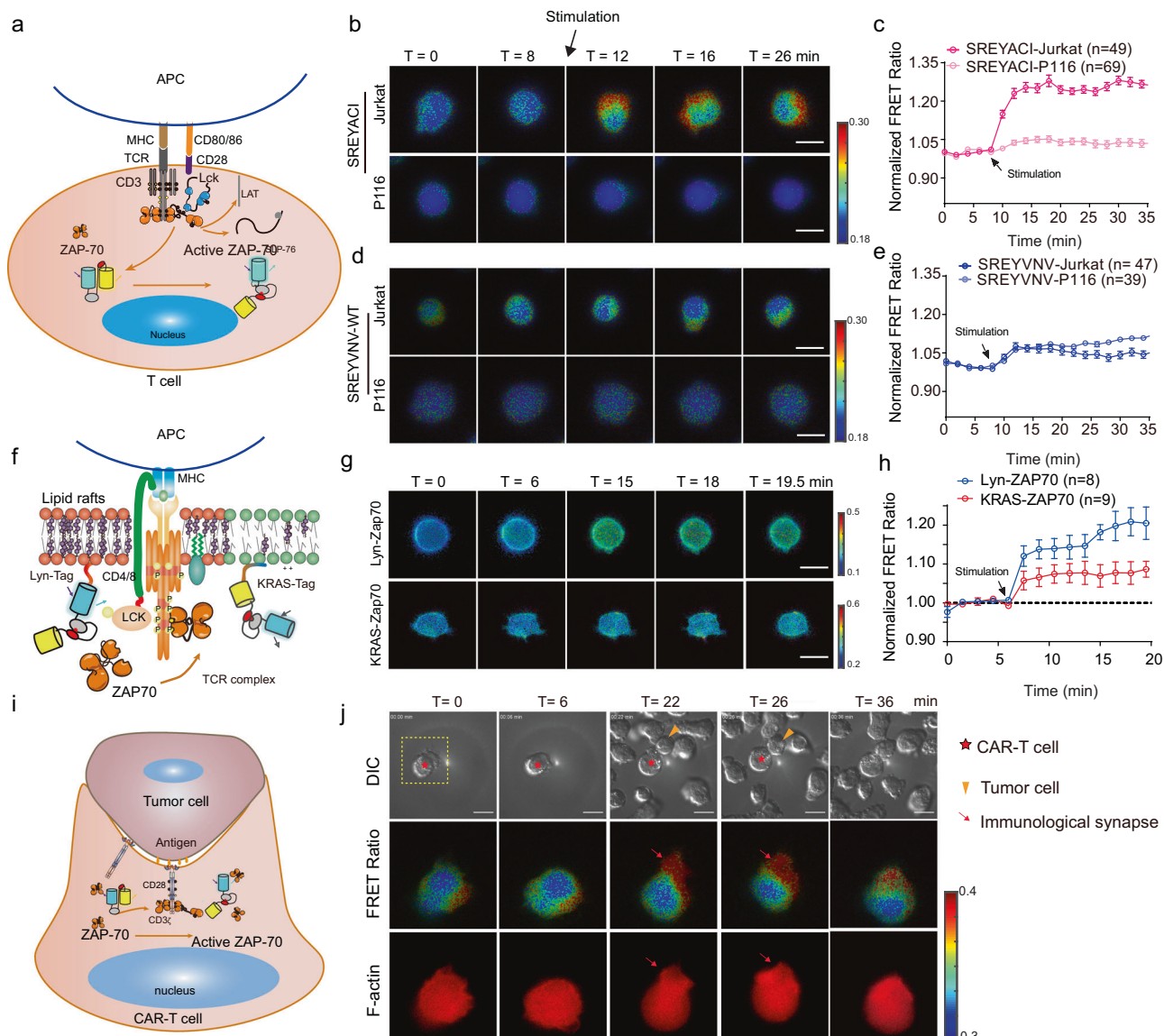

**Fig. 4 The sensitivity and specificity of the ZAP70 FRET biosensor in human T cell. a** Working mechanism of the ZAP70 FRET biosensor in reporting TCR signaling. **b–e** Time-lapse ECFP/FRET ratio (FRET ratio) images (**b, d**) and time courses (**c, e**) of improved (**b, c**) or parental (WT) (**d, e**) biosensors before and after TCR activation induced by CD3/CD28 antibody stimulation (*n* is shown in the figure). Error bars, mean ± SEM. Scale bars, 10 μm. See also supplementary video 5. **f** Schematics of membrane-bound biosensors which target different membrane compartments. Lyn- and Kras-ZAP70 FRET biosensors target the lipid rafts or non-raft regions, respectively. **g–h** Time-lapse FRET ratio images (**g**), and the normalized FRET ratio (**h**) of ZAP70 activities in different membrane compartments after TCR activation before and after CD3/CD28 antibody stimulation. (*n* is shown in the figure). Error bars, mean ± SEM. Scale bars, 10 μm in (**g. i**). Schematics of CD19-CAR Jurkat T cell engaging with a CD19+ tumor Toledo cell. **j** Time-lapse FRET ratio images of CAR-T cell expressing the improved ZAP70 FRET biosensor before and after the engagement with a target tumor Toledo cell. Each biological replicate had similar results (*n* = 4). Scale bars, 10 μm. See also supplementary video 6. The color bar in (**b, d, g, j**) indicates FRET ratio (ECFP/FRET), with hot and cold colors representing the high and low ratios, respectively. Source data are provided as a Source Data file.

XX3-CAR-T cells comparing to WT-CAR-T cells (Fig. 5b–g), suggesting a significant ZAP70 defect in XX3-CAR-T cells. To verify the results of single-cell imaging, we further evaluated ZAP70 activity in cells expressing different CARs in a large-scale, high-throughput manner using a flow cytometer with the optimized ZAP70 biosensor. Consistent with imaging experiments, a remarkable increase in the percentage of the ZAP70-active CAR-T cells (as indicated by high-FRET ratios) was observed after CD19+ Raji stimulation (Fig. 5h–i). There was a significantly higher proportion of ZAP70-active cells in WT-CAR-T cells than that of XX3-CAR-T cells (Fig. 5j–l), confirming that WT-CAR is more effective in activating ZAP70 than XX3-CAR. These results

suggest that ITAM motifs may affect ZAP70 to modulate CART cell functions. Our results also demonstrate that this optimized ZAP70 biosensor is applicable to monitor and evaluate the signaling of different synthetic CAR molecules and enables the high-throughput screening of functional/improved CARs by FACS.

**High throughput saFRET-based drug screening**. We reason that the saFRET biosensor design also provides a platform for biosensor-based HTDS in living cells. The carry-on-kinase-domain should bypass the need of maintaining endogenous regulation of the target kinase, minimize the heterogeneity and noise of endogenous kinase activation, avoid identifying upstream

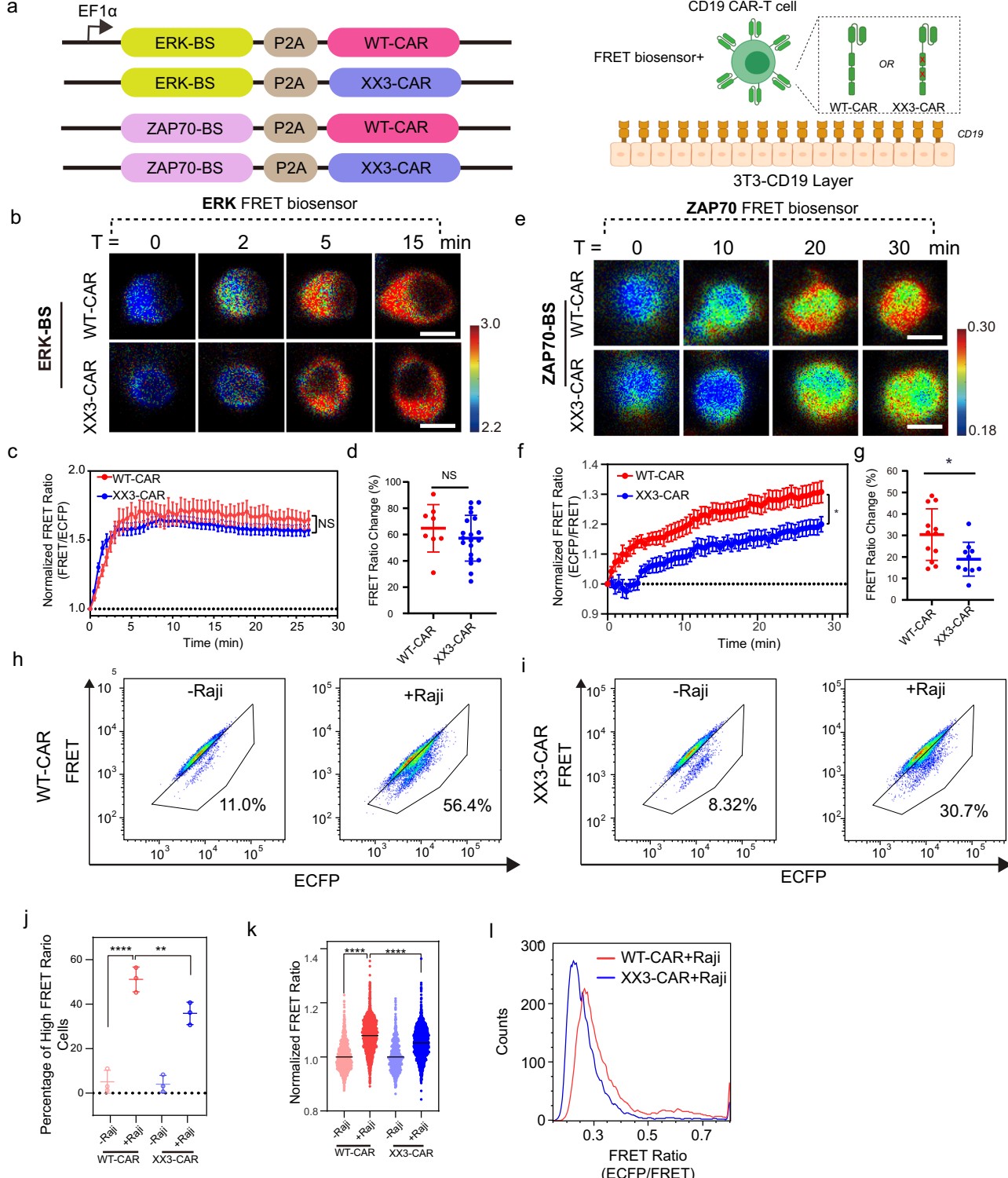

pathway regulators instead of on-target hits, and provide flexibility of choosing suitable cell systems that maintain live-cell contexts but offer experimental ease. In the case of ZAP70, this platform should enable us to screen small molecule inhibitors of ZAP70 kinase activity in adherent HEK cells, which overcomes the limitation of using suspension cells (Supplementary Fig. 10a) and is compatible with HTDS utilizing FRET imaging. Furthermore, HEK cells lack endogenous ZAP70 kinase[52] and hence have minimal heterogeneous background noise of individual

cells, in contrast to that of T cells. A stable HEK cell line was first established to express similar copy numbers of the ZAP70 saFRET biosensor (SREYACISGEL) whose average FRET ratio reduction was ~25% when treated with TAK695 (25 μM) without noticeable cytotoxicity (Supplementary Fig. 10b–d). This sensitivity of the biosensor should allow an image-based, high-throughput platform in 96-well glass-bottom plates capable of both endpoint and quantitative dynamic FRET measurements (Fig. 6a), thus overcoming the difficulty of using the parent

**Fig. 5 Less ZAP70 activation in XX3-CD19 CAR-T cell. a** Schematic drawings of constructs. The ERK- or ZAP70-FRET biosensor was co-expressed with WT-CAR or XX3-CAR in Jurkat T cells. In XX3-CAR, the tyrosine in the first and second ITAM motif was mutated to phenylalanine. The CAR-T cells were then dropped onto the 3T3 cells that constitutively express CD19 to monitor the dynamic ZAP70 or ERK kinase activations. **b** Representative images of ERK biosensor in T cells after attaching to the CD19[+]-3T3 cell monolayer. Scale bar,10 μm. **c** Time courses of FRET ratio (FRET/ECFP) of ERK FRET biosensor in WT- or XX3-CAR-T cells ($n = 8$ and 20 in each group). Error bars, mean ± SEM. **d** Percentage changes of ERK-FRET biosensor in WT- or XX3-CAR-T cells ($n = 8$ and 20 in each group, unpaired two-tailed Student's $t$-test, NS, $P > 0.05$). Error bars, mean ± SD. **e** Representative images of ZAP70 biosensor in T cells after attaching to the 3T3 CD19[+] cell monolayer. Scale bar = 10 μm. **f** Time courses of FRET ratio (ECFP/FRET) of ZAP70 biosensor in WT- or XX3-CAR-T cells ($n = 12$ and 10 in each group). Error bars, mean ± SEM. **g** Percentage changes of ZAP70-FRET biosensor in WT- or XX3-CAR-T cells ($n = 12$ and 10 in each group, unpaired two-tailed Student's $t$-test, $*P = 0.017$). Error bars, mean ± SD. **h–i** Flow-cytometry analysis of ECFP/FRET in WT-CAR-T cells (**h**) or XX3-CAR-T cells (**i**) before and after CD19[+] Raji cell stimulation. Each biological replicate had similar results ($n = 3$). **j** Percentage of High-FRET ratio cells in different groups of three independent experiments. ($n = 3$ in each group, One-way ANOVA, $****P < 0.0001$, $**P = 0.0054$). Error bars, mean ± SD. **k** Normalized FRET ratio of WT- or XX3 CAR-T cells before and after CD19[+] Raji cell stimulation. (From left to right, $n = 1314$, 3151, 682, and 3635, respectively, One-way ANOVA, $****P < 0.0001$). Error bars, mean ± SEM. **l** Histogram of FRET ratio in WT- or XX3-CAR-T cells after CD19[+] Raji cell stimulation. Source data including the gating strategies are provided as a Source Data file.

biosensor with low sensitivity and unsuitable for HTDS assays[33] (Supplementary Fig. 6b).

We first screened a 96-member kinase inhibitor library to identify efficient inhibitors of ZAP70 kinase. During the screening, library inhibitors at 10 μM were used as the screening dosage to identify the inhibitors more potent than TAK-659, which was relatively ineffective in suppressing ZAP70 activity at 10 μM (Supplementary Fig. 11a). After 40 min of incubation with inhibitors, the cells were imaged, and the FRET ratio changes of individual inhibitors compared to the solvent control were calculated to identify promising candidates after this primary endpoint screening (Fig. 6b–d). We further exploited a control biosensor with a kinase-dead domain (saFRETkd) for counter screening to eliminate false-positive hits due to auto-fluorescence or other non-specific effects[53]. Our results showed that the hits using the conventional FRET biosensor assay include inhibitors that target upstream signaling molecules, but not ZAP70 itself (Supplementary Fig. 10e–f). These false-positive hits were successfully eliminated in our saFRET-HTDS assay (Fig. 6c and supplementary Fig. 10g–h), thus demonstrating the specificity of our saFRET screening assays. Among the candidates identified, PHA665752, GSK2334470, and Pi3Kα inhibitor 2 showed a nonspecific reduction of FRET ratio with saFRETkd, which were eliminated from further analysis (Fig. 6e and Supplementary Fig. 10b). Dynamic tracking of the changes in FRET ratio of the remaining biosensor candidates after inhibitor treatment verified the endpoint screening results (Fig. 6f, g). This HTDS assay revealed that Staurosporine, AZD7762, and Sunitinib [an FDA approved drug for renal cell carcinoma (RCC) and imatinib-resistant gastrointestinal stromal tumor (GIST)] from the compound library can effectively inhibit the ZAP70 activity. While staurosporine can be toxic to cells, cell death was not observed after 30 min of staurosporine treatment with the dosage used in our experiments, when the FRET ratio was shown to be significantly reduced (Supplementary Fig. 11c–d). These results suggest that our saFRET platform can identify inhibitors targeting ZAP70, independent of cytotoxicity.

**Staurosporine, AZD7762, and Sunitinib inhibit clinical-relevant T cell activation.** We further tested the phosphorylation of LAT, a downstream substrate of ZAP70 kinase, and the activation of T cells with the identified inhibitors, staurosporine, AZD7762, and sunitinib (Fig. 6d). Significant reductions of phosphorylated LAT(Y191) (Fig. 7a–c) and ZAP70 (Fig. 7d and Supplementary Fig. 12) were observed after treatments with these inhibitors, verifying the efficacy of the biosensor screening results. These small molecules also suppressed the expression of CD69 (Fig. 7e, f), a surface marker representing activated T cells.

To examine the efficacy of staurosporine, AZD7762 and sunitinib in inhibiting pathological T cell activation, we used a disease model where T cell activation is mediated by ZAP70-R360P mutation, the main cause of a severe human autoimmune syndrome[4]. We introduced wild-type ZAP70 or the R360P-mutant of ZAP70 into ZAP70-deficient P116 T cells (Fig. 7g). ZAP70-R360P led to an enhanced T cell auto-activation compared to the wild-type ZAP70 (Fig. 7h). Staurosporine, AZD7762, or sunitinib significantly inhibited the phosphorylation of ZAP70 and LAT(Y191) (Fig. 7i–k and Supplementary Fig. 12), and the subsequent activation of T cells marked by CD69 (Fig. 7l and Supplementary Fig. 13).

## Discussion

We have developed the FRET-Seq platform to improve FRET biosensors directly in mammalian cells in a high-throughput manner. This strategy combines several techniques to improve FRET biosensor sensitivities by modifying and screening substrate sequences. First, high-throughput FACS screening and sorting based on FRET allow the selection of improved variants from libraries on a large scale, whose sequences can then be identified by the integration of NGS and analysis. Second, the saFRET design can overcome difficulties in mammalian-cell library screening caused by the heterogenic kinase activities from individual cells. Third, the counter-sorting strategy incorporating a kinase-active or kinase-dead domain in biosensor variants promotes the biosensor specificity during the screening process. While the focus of this work is on the substrate sequence of the biosensor, optimizations of linkers[54], SH2 domains[9], and FPs[3] may further enhance the performance of the biosensor and our FRET-Seq platform can be extended to screen more diverse libraries to optimize these important components of the FRET biosensors in the future. The integration with in silico simulation to identify the hot spots for mutagenesis should further increase the success rate of library screening[55]. While Fyn and ZAP70 kinase biosensors were chosen as the primary targets in this study to provide the proof-of-concept and verification for our approach, the FRET-Seq platform is modular and extendable to optimize other fluorescent biosensors, particularly those for detecting enzyme-based posttranslational modifications. These improved biosensors should enable us to monitor signaling events in single live cells with high sensitivity and specificity. It is of note that the performance of FRET biosensor is determined by multifactors (e.g., kinase selectivity of substrates, the orientation and affinity of the substrate binding to SH2 domain), the substrates that are optimal for biosensor hence may not be the same as the ones preferred only by the kinases[56,57].

The high-throughput FRET imaging platform using the improved saFRET biosensors allowed the HTDS of efficient and

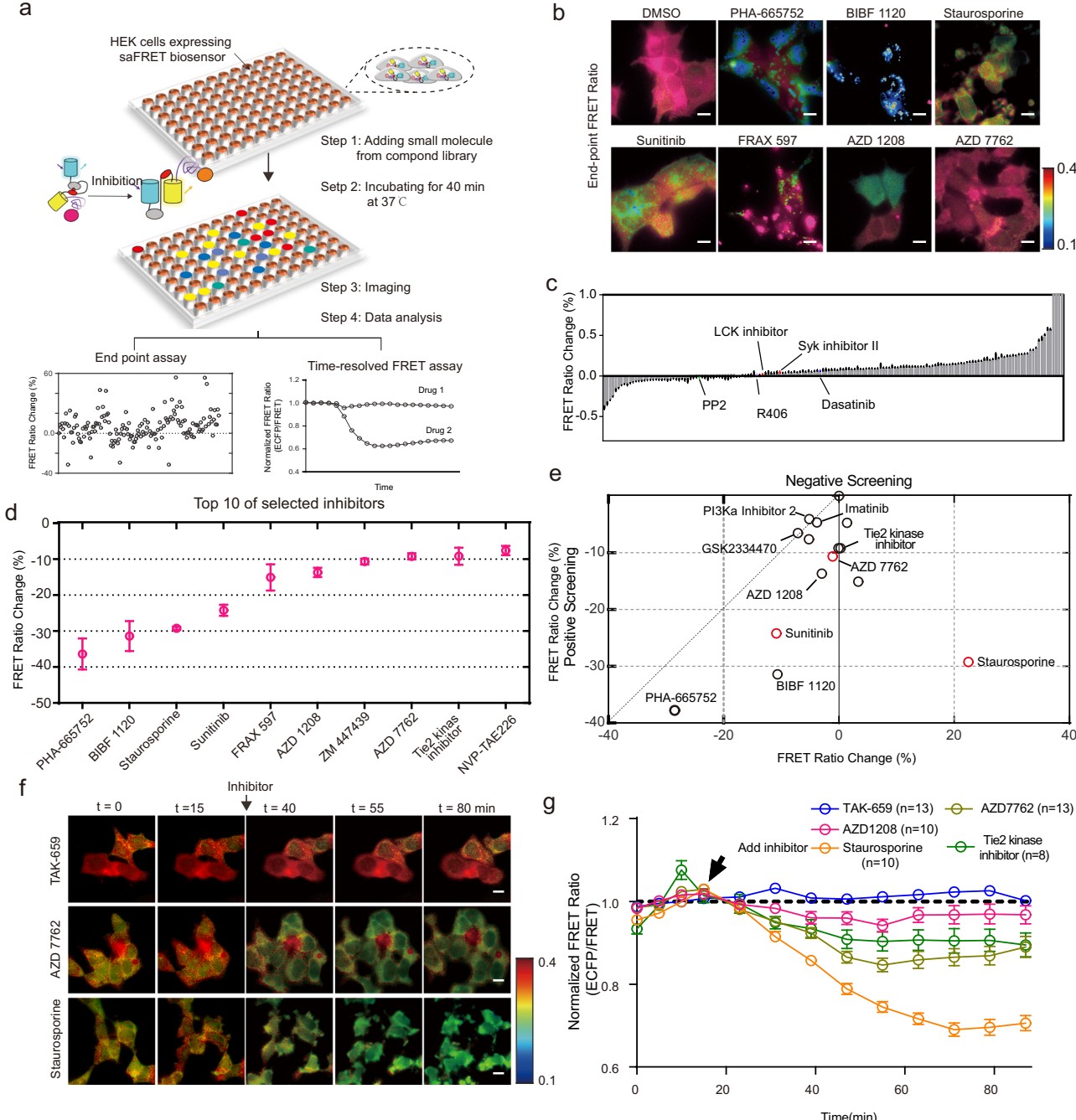

**Fig. 6 High-throughput drug screening platform using saFRET biosensor. a** Schematics of the high-throughput drug screening platform. First, the cells cultured in 96-well glass-bottom plates were treated either with DMSO or inhibitors from the kinase inhibitor library. After 40 min of incubation, the cells were imaged, and the FRET ratio change compared to the control cell was calculated. This platform can also allow dynamic tracking of the FRET ratio change after inhibitor treatment in single cells. **b** Representative FRET-Ratio images of the cells with different inhibitors corresponding to (**d**). Scale bars, 10 μm. **c** Summary of screening results. Some of the inhibitors have shown high efficiency in inhibiting ZAP70 kinase. The highlighted inhibitors represent the inhibitors targeting ZAP70 upstream signaling molecules. Dasatinib, Src kinase inhibitor; PP2, Lck/Fyn kinase inhibitor; R406, Syk inhibitor. **d** Top 10 selected inhibitors (From left to right, $n$ = 24, 12, 31, 37, 26, 50, 49, 51, 46, and 46, respectively). Error bars, mean ± SD. **e** Counter screening using a mutant biosensor with a kinase-dead domain to subtract the noise engendered from non-specific fluorescence. The Scatter plot illustrates the FRET ratio changes in the positive and negative screenings using the saFRET biosensor fused with an active kinase or a kinase-dead domain, respectively. **f–g** Time-lapse FRET ratio images (**f**), and the normalized FRET ratio (**g**) of HEK293 cells before and after inhibitor treatment. The TAK-659 (10 μM) was used as the negative control, which cannot sufficiently inhibit the ZAP70 kinase. ($n$ is shown in the figure). Error bars, mean ± SEM. Scale bars, 10 μm. See also supplementary video 7. Source data are provided as a Source Data file.

specific small molecules in live mammalian cells[29], overcoming issues in conventional assays related to cell permeability and cytotoxicity[58–60]. In fact, this saFRET-HTDS assay is different from the conventional FRET assays in multiple ways. Since

saFRET biosensor has the corresponding kinase domain fused to the FRET biosensor, this design enables us to screen small molecule inhibitors in adherent cells for kinases typically expressed in suspension cells, thus overcoming the limitation and

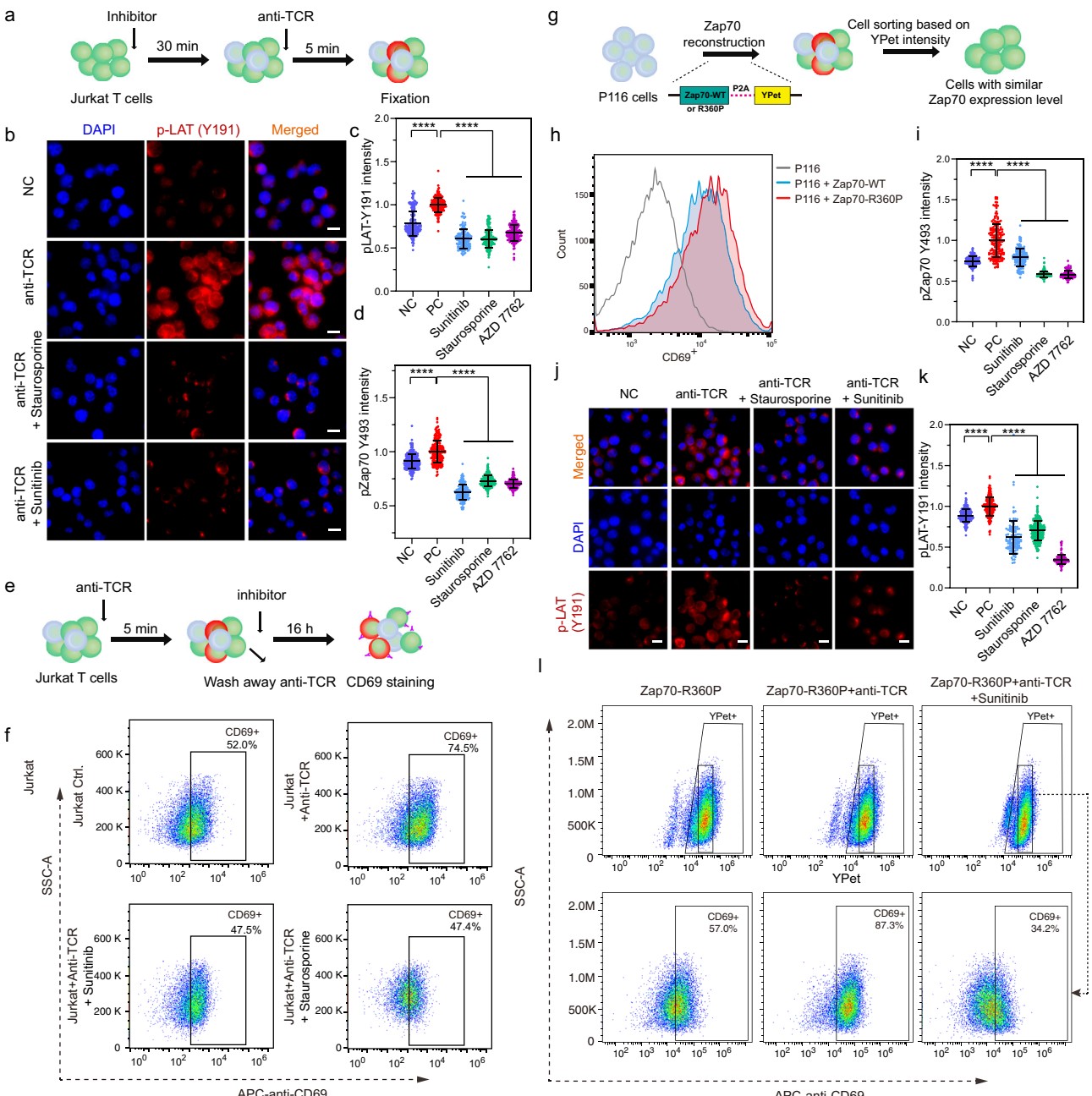

**Fig. 7 Inhibition of T cell activation by the HTDS-identified ZAP70 inhibitors. a** Experimental scheme and timeline for experiments in (**b**–**d**). Jurkat T cells were pre-treated with inhibitors for 30 min before anti-TCR stimulation by anti-CD3/CD28 antibodies for 5 min. **b** Immunostaining images of pLAT (Y191) in Jurkat T cells with different inhibitor pre-treatments. Scale bars, 10 μm. **c** Quantification of pLAT (Y191) intensity of single cells in different groups. (From left to right, $n = 197, 182, 139, 169, 213$, respectively, One-way ANOVA, ****$P < 0.0001$). NC represents Jurkat T cells without any treatment, PC represents Jurkat T cells stimulated with anti-TCR only. Data were normalized to the PC group. Error bars, mean ± SD. **d** Quantification of Phosphorylated ZAP70 (Y493) intensity of single cells in different groups. (From left to right, $n = 249, 222, 136, 276$, and 230, respectively, One-way ANOVA, ****$P < 0.0001$). Error bars, mean ± SD. **e** Experimental scheme, and timeline for CD69 staining experiment. **f** Flow-cytometry analysis of CD69 expression in T cells after anti-TCR stimulation, with or without inhibitors pre-treatment. Each biological replicate had similar results ($n = 3$). **g** Experimental scheme, and timeline of P116 cells reconstituted with ZAP70. Full-length ZAP70-WT or R360P were expressed with YPet via a cleavable P2A linker. P116 cells with similar ZAP70-WT or ZAP70-R360P expressions were sorted and isolated for further analysis based on YPet intensity. **h** CD69 expression in P116 cells with or without the expression of ZAP70 (WT) and its mutant (R360P). **i** Quantification of pZAP70 (Y493) intensity of P116-ZAP70 R360P cells with different inhibitor pre-treatments. (From left to right, $n = 202, 191, 150, 280, 110$, respectively, One-way ANOVA, ****$P < 0.0001$). Error bars, mean ± SD. **j** Images of pLAT (Y191) in P116-ZAP70 R360P cells with different inhibitor pre-treatments. Scale bars, 10 μm. **k** Quantification of pLAT (Y191) intensity of P116-ZAP70 R360P cells with different inhibitor pre-treatments. (From left to right, $n = 275, 252, 116, 240, 190$, respectively, One-way ANOVA, ****$P < 0.0001$). Error bars, mean ± SD. **l** Flow-cytometry analysis of CD69 expression in P116–ZAP70-R360P cells after anti-TCR stimulation, with or without sunitinib pre-treatment. ZAP70-R360P expression levels in P116 cells were indicated by YPet intensity. Source data including the gating strategies are provided as a Source Data file.

difficulty of imaging in suspension cells with the conventional FRET biosensors (Supplementary Fig. 10a). Besides, compared to conventional assays, saFRET-HTDS enables the screening of inhibitors directly targeting ZAP70 kinase, thus avoiding the false positive selection of inhibitors targeting upstream molecules of ZAP70 in host cells. Since ZAP70 kinase is crucial for T-cell functions, but can be compensated by Syk in innate immunity[61], specific inhibitors of ZAP70 kinase (but not targeting Syk) should not cause perturbation of innate immunity and hence can have selectivity in targeting T-cell related diseases, e.g., controlling allograft rejection and autoimmune diseases such as rheumatoid arthritis, and multiple sclerosis[24]. Furthermore, our saFRET biosensor with a ZAP70 kinase-dead domain in HEK cells expressing Syk kinase[62] had a low basal level, and did not show any response after TAK-659 treatment which targets both ZAP70 and Syk (Fig. 3b, c). These results suggest that the FRET signal of our saFRET biosensor is unlikely affected by the Syk activities. Thus, the inhibitors identified using this saFRET assay should be selective for ZAP70 kinase. This is significant as the inhibition of both ZAP70 and Syk can have devastating effect on immunity and coagulation. The identification of staurosporine and AZD7762 as ZAP70 inhibitors, despite their potential cytotoxicity in long terms and limited therapeutic potentials, demonstrated the successful application of the saFRET-HTDS platform in identifying ZAP70 inhibitors; further modifications of these inhibitors (e.g., conjugated to liposomes nanoparticles[63] or gold nanoparticles[64]) could potentially enhance the specificity and reduce the cytotoxicity in the future. It should be noticed that sunitinib identified in this study was reported to reduce the levels of phosphorylation of ZAP70, but did not affect the activation of Lck, the direct upstream effector of ZAP70[65]. Together with our results of saFRET assays, sunitinib can be a relatively specific ZAP70 kinase inhibitor. Since sunitinib is an FDA-approved drug whose safety has been demonstrated in human subjects, we expect that sunitinib has high therapeutic potentials in treating ZAP70 related autoimmune diseases [e.g., hyperactive ZAP70 R360P mutation-mediated autoimmune disease in patients needing allogeneic hematopoietic cell transplantation (HCT)[4]]. Using high content screening platforms equipped with fully automated cellular imaging apparatus and analysis algorithms, the saFRET-HTDS system should also be applicable to screen large-scale compound libraries for drug discovery or repurposing of FDA-approved drugs[66]. These large-scale libraries can also allow counter-screening against other kinases (e.g., Syk, Lck) incorporating into our saFRET biosensors to further screen ZAP70 inhibitors with high selectivity.

## Methods

**Cell culture and transfection**. Human embryonic kidney cells (HEK 293T) were cultured in Dulbecco's Modified Eagle's Medium (DMEM) supplemented with 10% (v/v) fetal bovine serum (FBS), 2 mM of L-glutamine, 100 units/mL of penicillin, 100 μg/mL of streptomycin, and 1 mM of sodium pyruvate at 37 °C with 5% $CO_2$. All the cell culture reagents were purchased from Thermo Fisher Scientific. The plasmids were transfected into cells with Lipofectamine 3000 (Thermo Fisher Scientific, Cat. No. L3000015). Jurkat and P116 cells were cultured in RPMI 1640 medium (Gibco) supplemented with 10% FBS, 1% penicillin-streptomycin, 1% sodium pyruvate, and L-glutamine (300 mg/L). Human PBMCs were isolated from buffy coats (San Diego Blood Bank) after donor de-identification using Ficoll gradients (Amersham Biosciences). Human primary CD4+ T cells were isolated from PBMCs using CD4+ T cell isolation kit (Miltenyi). For lentiviral transduction, isolated CD4+ T cells were activated for 72 h in complete RPMI medium supplemented with phytohemagglutinin (PHA, Fisher Scientific, R30852801) and IL-2 (100 IU/mL). The cells were then infected with concentrated lentivirus at an MOI (multiplicity of infection) of 10 by spinoculation on Retronectin (Takara)–coated plates.

**Plasmid construction**. A step-by-step protocol describing the FRET-Seq assay can be found at Protocol Exchange[67].

The gene template for the protein tyrosine kinase biosensor was constructed by polymerase chain reaction (PCR) amplification of the complementary DNA of an Enhanced CFP (ECFP), LacZ, YPet, EV linker (116 amino acids), and either active or dead kinase domains. The amplified gene elements were cloned into the pSin lentiviral transfer vector (pSin-ELYK, where E, L, Y, and K stand for ECFP, LacZ, YPet, and kinase domain, respectively), between SpeI and EcoRI with T4 ligation (New England Biolabs, NEB) where ECFP is at N-terminus and the kinase domain is at C-terminus. Several restriction sites were introduced: two Esp3I sites at each end of LacZ to be replaced by the sensing domain (including SH2 domain, a flexible linker (15 amino acids), and substrate peptide), and XbaI/EcoRI flanking the kinase domain to be replaced by different domain versions. To construct the biosensor, the cDNA of the sensing domain, which was amplified by PCR from the mutated c-Src SH2 domain (C185A)[9] with a sense primer containing an Esp3I and a reverse primer containing the cDNA of a flexible linker, a substrate peptide, and an Esp3I site, was used to replace the LacZ domain via the Golden Gate assembly (NEB). To vary the sequence of the substrate peptide, the PCR products of a common forward primer and a NNK degenerate primer that target the substrate were inserted into the ELYK template by Golden Gate assembly. Since the regulation of phosphorylation level in HEK293 cells depends on the interaction between the substrate and the kinase domain, we tested several kinase domains with different lengths for their capability in phosphorylating substrate in biosensors using western blotting. In western blotting, the Anti-GFP antibody (Abcam, Cat. No. ab290, 1:5000 dilution) was used to target the biosensor, and Anti-Phosphotyrosine antibody (clone 4G10, Upstate, 1 μg/ml working solution) was used to target the phosphotyrosine. To identify an optimized kinase domain for saFRET biosensor, Fyn kinase domains ranging from 265-526 and ZAP70 kinase domains ranging from 327-619, 327-601 were tested. A K299M or K369A mutation was introduced into the Fyn and ZAP70 kinase domain, respectively, to generate the kinase-dead control. The well-established substrate (EKIEGTYGVV) for Fyn biosensor was chosen for the Fyn biosensor[35]. The previously published substrates for ZAP70 kinase FRET biosensors from Vav2, LATY175, and LATY191 were tested using Western blotting. The biosensors with selected kinase domain and substrate were used as a starting template to generate biosensor libraries.

**Biosensor library construction**. There were four libraries for one kinase biosensor optimization, including two libraries for the biosensor with active kinase domain (KA) and two libraries for the biosensor with dead kinase domain (KD). The biosensor libraries with KA or KD can be further separated into two groups, including, Lib1 and Lib2, which represent the substrate peptide libraries whose three amino acid residues upstream and downstream of the consensus tyrosine residue of the wild-type substrate were subjected to site-saturation mutagenesis, respectively. NNK degenerate primers (Integrated DNA Technologies, IDT) were used to generate these libraries, where N represents an equimolar distribution of A, T, G, and C; K represents an equimolar distribution of T and G; X represents any amino acid. Briefly, three primers were used to generate the substrate library of Lib1 and Lib2 for one type of the kinase domain. Thus, 32768 (NNKNNKNNK = 32*32*32) DNA sequences which correspond to 8000 amino acid sequences (XXX = 20*20*20) were generated. The cDNA of the substrate variants was generated by PCR with Q5 DNA polymerase (NEB, Cat. No. M0491) from the c-Src SH2 domain (C185A) with a sense primer containing an Esp3I (Primer#1), and the antisense primers containing NNK codons, which were labeled as Primer#2 for Fyn Lib-1 and Primer#3 for Fyn Lib2. For ZAP70-biosensor library, the sense primer was the same as Primer#1, and the antisense primers were labeled as Primer#4 for ZAP70 Lib-1 and Primer#5 for ZAP70 Lib-2. All the sequences of these primers were listed in Supplementary Data 7. The template for PCR was the wild-type substrate. The PCR condition of annealing temperature was varied from 55 to 70 °C and thermocycling condition was 20 cycles. After generating the substrate mixture with NNK degenerated reverse primers, the fragments containing cDNA mixture were then extracted from agarose gel, inserted into pSin-ELYK template vector between Esp3I restriction sites through the Golden Gate assembly (NEB) to generate the biosensor library. The product was purified and concentrated by using DNA Clean and Concentrator Kits (Zymo Research) and transformed into ElectroMAX™ DH10B™ Cells (Invitrogen, Catalog number: 18290015), which are electrocompetent E. coli cells offering the highest transformation efficiencies of >1 × 10^10 CFU/μg plasmid DNA. Then the plasmids containing libraries were purified with Qiagen HiSpeed Plasmid Maxi kit (Qiagen) and the mutation regions of the libraries were further verified using Sanger sequencing.

**Generation of mammalian cell library**. The plasmids of biosensor libraries were introduced into mammalian cells (HEK293T cells from ATCC) through virus infection with low MOI (0.1) to allow low copy number of plasmids per single cell. Lentiviruses were produced from Lenti-X 293T cells (Clontech Laboratories, #632180) co-transfected with a pSin containing biosensor variants and the viral packaging plasmids pCMV-△8.9 and pCMV-VSVG using the ProFection Mammalian Transfection System (Promega, Cat. No. E1200). Viral medium/supernatant was collected 48 h after transfection, filtered with 0.45 μm filter (Sigma-Millipore), and concentrated using PEG-it virus precipitation solution

(System Biosciences, Cat. # LV825A-1). The virus titer was measured by flow cytometry. To generate the mammalian cell library, we added the concentrated virus with the MOI of 0.1 into HEK293T cells, which were seeded with a density of $2 \times 10^6$ cells in a 10-cm dish a day before transfection. In this case, ~95% of the infected cells in our experiments have only one variant, which can be assessed by the following analysis: the ratios of functional versus target cells can be described in terms of Poisson distributions[68]; thus, the percentage of the non-infected cells can be calculated by $P(0) = e^{-0.1} = 90.48\%$, and the percentage of cells infected with only one virus is $P(1) = 0.1e^{-0.1} = 9.05\%$; the remaining 0.47% cells have more than one variant (See also: https://www.virology.ws/2011/01/13/multiplicity-of-infection); we eliminated the non-infected cells via FACS; as a result, ~95% of the infected cells have only one variant in the sorted cells which were subjected to further experiments. The enrichment ratio was used to rank the variants and counter sorting was further used to eliminate the noises engendered during screening and sequencing.

**Mammalian cell library screening by FACS.** HEK 293T cells containing biosensor variants were screened by FACS (BD FACS Aria II Cell Sorter) according to FRET ratio (ECFP/FRET ratio). This ratio is calculated by dividing the ECFP emission intensity over the FRET signal intensity. Several controls were utilized to gate the desired cell populations. Plain HEK 293T cells were used to gate for the fluorescence channels. ECFP- or YPet-expressing cells were used to gate for cells that expressed ECFP only (Ex 405 nm, Em 450/50 nm) or YPet only (Ex 488 nm, Em 545/35 nm), respectively. The mixture of cells that express either ECFP or YPet was used as a negative gating of FRET signal (Ex 405 nm, Em 545/35 nm). Cells co-transfected with both ECFP and YPet were also used to gate for the intermolecular FRET signal. The conformations of intramolecular FRET biosensors can be measured based on the FRET ratio. Thus, cells expressing biosensors (with the wild-type substrate) fused with active kinase domain (KA) were used to gate for the active conformation of FRET biosensor (high FRET ratio), while those expressing biosensors (with the wild-type substrate) fused with dead kinase domain (KD) were used to gate for the inactive conformation of FRET biosensor (low FRET ratio). In addition to all gate settings, only cells expressing the medium intensity of biosensor (by YPet filter; Ex 488 nm, Em 545/35 nm) were selected for sorting to avoid the abnormal expression of biosensors in cells and the consequent noises. After the gate setting, cells containing biosensor libraries were sorted into high and low FRET ratio, where the median FRET ratio was used as a threshold for sorting. Top ~5% of the cells expressing biosensors with a higher FRET ratio than the median FRET ratio were selected for the high-FRET ratio and vice versa for the low-FRET ratio. Based on the FRET ratio, the cells were sorted into four groups for each library, such as, KAH (High FRET ratio with Active Kinase, KA), KDL (Low FRET ratio with Dead Kinase, KD), KAL (Low FRET ratio with Active Kinase, KA) and KDH (High FRET ratio with Dead Kinase, KD). In addition to the sorted cells, the cells collected prior to sorting in each library were kept as the input control for high-throughput sequencing. All the FACS related data were analyzed using FlowJo software 10 (BD).

**Illumina DNA sequencing of biosensor variants.** The substrates of selected biosensor libraries were sequenced by the Illumina HiSeq 4000 sequencing system. After sorting, the total RNA of each pool of sorted cells and non-sorted cells was extracted by RNeasy Mini Kit (Qiagen, Cat# 74104). During column purification, the genomic DNA was removed by RQ1 RNase-Free DNase (Promega, Cat# M6101). This allows only RNAs encoding the biosensor proteins to be purified. The RNAs were then quantified by Nanodrop and gel electrophoresis. The purified total RNA (~500 ng) was used as a template for cDNA synthesis via the SuperScript IV reverse transcriptase (Thermo Fisher Scientific, Cat# 18090010) with gene-specific primer (Supplementary Data 7). Adaptor sequences with different indexes for Illumina sequencing were added into cDNA by PCR using Q5 DNA polymerase (NEB, Cat# M0491S) with low PCR cycles (<16 cycles). Illumina sequencing fusion primers were synthesized from IDT. To generate sequencing libraries, the same strategy was used for Fyn and ZAP70 libraries. Taken the Fyn-biosensor Library as an example, the forward primer for library sequencing was labeled as Primer#6, which contains the Illumina's flow cell binding sequence, sequencing primer sites, and constant regions in the substrate sequence that are not mutated. The reverse primer was labeled as Primer# 7 which contains the Illumina's flow cell binding sequence, sequencing primer sites, adaptor, and constant regions in the substrate sequence that are not mutated (Sequence of these primers can be found in the Supplementary Data 7). The individual pools of the libraries were labeled with different barcodes by adjusting the adaptor sequence of the reverse primers (Primers#8–22, Supplementary Data 7). The library size of amplicons containing all adaptors was confirmed by gel electrophoresis (2% agarose gel) and the amplicons were purified by Zymoclean gel DNA recovery kit (Zymo Research, Cat# D4008). The purified amplicon libraries were then sequenced by Sanger sequencing (Genewiz) to verify the success of library preparation and quantified by Qubit prior to being sequenced by Illumina HiSeq4000 with 50-bp single-end sequencing (for the entire libraries).

**Analysis of sequencing results and selection of biosensors with high sensitivity.** Sequencing data were analyzed using the Matlab (MathWorks) using

custom-written code. Only the sequences which had Phred score >20 at all positions covering the constant regions of the substrate sequence and TAC encoding tyrosine, and had the correct length of the insert were selected and converted from nucleotide sequence to amino acid sequence. Mutseq Version V1.10b was used to analyze the library sequencing data and the source code is available on a Github site: https://github.com/jason8301/mutseq[69]. To avoid bias due to sequencing depths, the frequency of unique sequences ($f_v$) was computed by normalizing the variant count in each library group to the total number of sequencing reads for that group. The reading counts of each sequence were normalized to Counts Per Million (CPM) to represent the frequency of a unique sequence. Sequences with CPM > 10 were considered positive and selected for further analysis. Different libraries had different total sequencing reads. Thus, the frequency of a unique sequence ($f_v$) can be compared across different library groups. The change in frequency of each variant from input to the selected groups could serve as a measure of its enrichment toward the selection criteria because following the counter-sorting criteria, FACS should enrich cells containing functional variants satisfying the selection criteria but deplete cells containing nonfunctional variants. These frequency data were later used to compute variant enrichment ratios ($E_v$), which allowed us to find the fold enrichment of that variant before ($f_{v,input}$) and after ($f_v$) sorting. In summary, for each variant, $v$, we can determine the frequency, $f_v$, by normalizing the total reads of the variant by the total reads of all variants in the library according to

$$f_v = \frac{c_v}{\sum_{k=1}^{n} c_k} \qquad (1)$$

where $n$ represents the total number of unique variants in a sequenced library and $c$ represents the total number of occurrences of a single variant denoted by the subscript. This normalization reduces the bias due to cell number and biosensor expression differences amongst different sorted groups.

The frequency of a single variant can then be compared between groups in calculating an enrichment ratio[41]. The enrichment ratio, $E_v$, for any single variant, $v$, is calculated by dividing the frequency of the variant from the selected library, $f_{v,sorted}$, over that from the input control, $f_{v,input}$, according to

$$E_v = \frac{f_{v,sorted}}{f_{v,input}}. \qquad (2)$$

A biosensor with better performance accurately responding to the kinase domain should be enriched in both KAH (High FRET ratio with Active Kinase, KA) and KDL (Low FRET ratio with Dead Kinase, KD) groups, meanwhile, it should not be enriched in KAL (Low FRET ratio with Active Kinase, KA) or KDH (High FRET ratio with Dead Kinase, KD) group. Therefore, the variants with $E_v > 1$ in KAH and KDL groups and <1 in KAL and KDH groups were selected. The data for each substrate sequence were visualized in the 4D plot using Matlab software. To achieve a better illustration of the 4D-plot, $E_v$ of each group was further normalized to $E_{vn}$ by $E_{vn} = \log (E_v)$ when $E_v > 1$, and $E_{vn} = E_v - 1$ when $E_v \leq 1$. These selected sequences with 4D analysis were further evaluated by their product of $E_v$ (KAH) and $E_v$ (KDL) and calibrated with the product of WT biosensor to filter and identify biosensors with the best performance.

**Microscopy, image acquisition, and quantification.** To verify the efficiency of the biosensor with the predicted substrate using fluorescent microscope, the substrates selected from Illumina's sequencing data of biosensor libraries were inserted into pSin-ELYK template to construct the biosensors (as described in Plasmid construction). After verifying the constructs by Sanger sequencing, the individual biosensor plasmid was transfected into cell using Lipofectamine 3000 (Thermo Fisher Scientific). Cells expressing the exogenous biosensor proteins were starved with 0.5% FBS DMEM for 12 h before being subjected to PP1 (10 μg/mL) or TAK-659 (25 μM) stimulation. In the PP1 wash out the experiment, the medium contains PP1 was replaced with a normal medium after wash three times. Images were taken with a Nikon Eclipse Ti inverted microscope with a cooled charge-coupled device (CCD) camera with a 420DF20 excitation filter, a 450DRLP dichroic mirror, and two emission filters controlled by a filter changer (480DF30 for ECFP and 535DF35 for YPet). The time-lapse fluorescence images were acquired by Meta-Morph 7.8 and the endpoint fluorescence images were acquired by MetaFluor 7.8 (Molecular Devices). The ECFP/FRET ratio images were calculated and visualized with the intensity modified display (IMD) method by Fluocell software[70] (Version: V6.0.0 Github http://github.com/lu6007/fluocell). For data presentation, the normalized values were shown to compare the differences among the experimental groups and to minimize the cell-cell heterogeneity. The pre-stimulation baseline for each cell was established by averaging the FRET ratio of each cell before stimulation.

**Cell membrane targeting biosensor.** A Lyn tag was added to the N-terminus of the ZAP70 biosensor through Gibson assembly (NEB). N-terminal glycine and cysteine in the acylation sequences can undergo myristoylation and palmitoylation, which leads to the specific targeting of the biosensor to membrane regions with saturated lipids where rafts micro-domain was formed after TCR activation[51]. The KRAS-ZAP70 biosensor was developed by adding the prenylation sequences (KKKKKKSKTKCVIM) to the C-terminus of the ZAP70 biosensor using Gibson assembly. C-terminal cysteine residue prenylation and

the neighboring polybasic amino acids can target the biosensor to the non-raft regions with non-saturated lipids.

**TCR and CAR signaling activation**. TCR signaling was activated by using CD3/CD28 antibody clusters[47] (Fig. 4) or anti-TCR[4] (clone C305, Sigma-Aldrich, Fig. 7) as described previously. Briefly, biotin-conjugated IgGs were mixed with CD3 antibody (10 µg/ml) and CD28 antibody (5 µg/ml) and were further clustered with streptavidin. Then the antibody clusters were used to stimulate the T cells cultured on nonspecific immunoglobulin G (IgG,10 µg/ml) coated glass-bottom dish. For co-expression of biosensor and CAR molecule, the wild-type or ITAM mutated (XX3) 1928ζ CAR was linked downstream of the gene of biosensor by a P2A linker in the lentiviral construct. The CD19 CAR-T cells expressing biosensors were generated by lentivirus transduction of Jurkat T cells. For imaging of CAR signaling upon antigen stimulation, CAR-T cells were dropped on the glass-bottom dishes that have been coated with the NIH-3T3 cells expressing CD19 antigens. The time-lapse fluorescence images were taken with a Nikon Eclipse Ti inverted microscope at an interval of 30 s. The W-VIEW GEMINI imaging splitting optics (Hamamatsu, Japan) with an iXon Ultra 897 camera was used to capture the ECFP (a 474/40 nm emission filter) and FRET (a 535/25 nm emission filter) fluorescent signals simultaneously. During Imaging, the cells were maintained with 5% $CO_2$ at 37 °C using the Tokai Hit ST Series Stage Top Incubator (Tokai Hit, Japan).

**Detecting ZAP70 activity by FACS**. The Jurkat T cells were first infected with the ZAP70 lentiviral vectors. Then, The Jurkat T cell line stably expressing ZAP70 biosensor (Jurkat-ZAP70) was established by cell sorting using Beckman moflo Astrios EQ cell sorter (Beckman coulter, CA). To produce CAR-T cells, the lentiviral construct co-expressing CAR and mCherry fluorescence gene was introduced into the Jurkat-ZAP70 cells. For CAR-T cell activation, $5 \times 10^5$ CAR-T cells were incubated with Raji cells at a ratio of 1:1 for one hour in 100 µl RPMI1640 medium at 37 °C. The CAR-T cells incubated without Raji cells served as a control. The cells were then fixed with 4% paraformaldehyde (PFA) for 10 min. Finally, the ZAP70 activity of CAR-T cells was detected by the flow cytometer (CytoFLEX LX, Beckman coulter, CA).

**Drug screening using saFRET biosensor**. Stable HEK293T cell lines expressing saFRET biosensors were established by sorting the cells with a similar biosensor expression level by FACS (Sony, SH800). Then 10,000 cells per well were cultured in the 96-well glass-bottom dish (Dot Scientific, MGB096-1-2-LG-L) for 12 h before drug testing. The glass-bottom dish was cleaned with 5% Hellmanex III (Sigma, Z805939) at 50 °C overnight following extensive washing with ddH$_2$O. After that, the dish was further cleaned with 5 M NaOH for 1 h following extensive washing with ddH$_2$O. Then, the well was dried using airflow and kept at room temperature after sterilization under UV light. For endpoint measurement, the fluorescence images were taken under microscope after 40 min of inhibitors (10 µM) from the kinase inhibitor library (Item No. 10505, Cayman) or DMSO treatment, before the FRET ratios were calculated as the primary screening. A control saFRET biosensor with a mutated dead kinase domain (K369A) was constructed and used in a counter screening to verify the selected inhibitors and eliminate the false-positive inhibitors which caused auto-fluorescence or non-specific effects. For secondary screening of short-listed inhibitors, the cells were seeded and treated with the same procedure as the primary screening. Instead of capturing the endpoint images as in the primary screening, live-cell imaging was performed and time course FRET ratio change of the cells was obtained and quantified. Only the top inhibitors identified from the primary screening were further tested, and these inhibitors that lead to a significant reduction of the FRET ratios in the secondary counter screening were eliminated from further analysis. To compare the conventional FRET assay, Jurkat T cells expressing the FRET biosensor were used. The T cells were cultured on nonspecific immunoglobulin G (IgG,10 µg/ml) coated glass bottom dishes, and the TCR signaling was activated by using CD3/CD28 antibody clusters after pretreatment with inhibitors for 30 min. The FRET ratio was automatically calculated by Fluocell software.

**Immunofluorescence staining**. T cells were cultured in donkey anti-goat IgG coated glass and starved for 30 min with inhibitor pretreatment before anti-TCR stimulation for 5 min. For the negative control group, the cells were starved for 30 min and treated with solvent DMSO. Then the cells were fixed in 4% paraformaldehyde for 15 min at 37 °C. After that, the cells were permeabilized with 0.1% Triton X-100 for 15 min and then blocked with 2% BSA for 1 h. Primary antibodies (phospho-ZAP70 (Tyr493)/Syk (Tyr526) (CST, #2704, 1:200 dilution) and phospho-LAT (Tyr191) (Invitrogen, #MA5-33177, 1:100 dilution) were diluted as recommended in blocking buffer and incubated with cells overnight at 4 °C. After three 10-min washes with PBS for three times, samples were incubated with the corresponding secondary antibodies, i.e., Alexa Fluor-594 donkey anti-Rabbit IgG (Thermo Fisher Scientific, #A21207, 1:1000 dilution) at room temperature for 1 h. After washes in PBS three times for 10 min, samples were counterstained with DAPI. The result was imaged subsequently with a Nikon Microscope. The intensity of the cells was quantified using Fluocell software

automatically. In staining experiments to measure T cell activation, CD69 antibody (BioLegend, APC anti-human CD69 Antibody, clone FN50, 1:100 dilution) was used to label the live cells for 30 min at room temperature followed by PBS washing for three times, then the cells were analyzed by flow cytometry (Sony, SH800).

**Statistics and reproducibility**. All statistical analyses were performed using GraphPad Prism version 9 or Matlab R2019b. For Library screening of improved FRET biosensor, we have reproduced the experiment twice for different kinases (i.e., Fyn and ZAP70 kinase) to ensure reliability and reproducibility. Moreover, the improved biosensors were further tested in at least three biological replicates and successful. Replication attempts were also successful for other experiments. Pairwise comparisons were performed using Student's $t$-test, and comparisons between more than two groups were performed using one-way ANOVA followed by Dunnett's multiple comparisons test to compare individual means as indicated in the figure legends. The sample size, statistical significance value and error bar graphs were listed in figure legends.

**Reporting summary**. Further information on research design is available in the Nature Research Reporting Summary linked to this article.

## Data availability
The library screening data are provided in the Supplementary Data set. All the other data is available in the main text and supplementary materials. The sequences of improved biosensors generated in this study have been deposited in the NCBI GenBank under accession code MZ542461 and MZ542462 for ZAP70-FRET(ACI) biosensor and saFRET-ZAP70(ACI) biosensor, respectively. Mutseq was used to analyze the next-generation sequencing data of the library[69]. The source code is available at a Github site: https://github.com/jason8301/mutseq. FluoCell[70] was used to analyze the FRET imaging data and the source code is available at a GitHub site: http://github.com/lu6007/fluocell. Source data are provided with this paper.

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

## Acknowledgements

We thank Professor Arthur Weiss at UC San Francisco for insightful suggestions and Drs. Enfu Hui and Takeya Masubuchi at UC San Diego for the help of T cell activation experiments. This work was supported in part by grants from NIH HL121365, GM125379, GM126016, R01EB03150, and CA204704 (Y.W.), DP1DK126138 (S.Z.), NSFC 31971324 (J.S.), and Zhejiang Provincial NSF LR20H160003 (J.S.).

## Author contributions

L.L., P.L., Y.H., J.S. and Y.W. designed research; L.L., P.L., X.M., Y.H. performed research; L.L., P.L., Y.H., R.H., T.H., Y.S., Y.Y., K.C. and S.L. analyzed data; L.L., P.L., S.Z., S.L. J.S., J.Z., S.C. and Y.W. wrote the manuscript. All authors reviewed the manuscript and have given approval to the final version of the manuscript.

## Competing interests

Y.W. is a scientific co-founder of Cell E&G Inc and Acoustic Cell Therapy Inc. These financial interests do not affect the design, conduct or reporting of this research. All the other authors declare no competing interests.

**Additional information**

