## [Peer Review File · Nature Communications]

Reviewers' Comments:

Reviewer #1:

Remarks to the Author:

REVIEW

In general:

The attempt to establish a platform to screen large-scale biosensor libraries in mammalian cells for cellular imaging and drug screening is not a simple matter and it has called for the attention of academic researchers. In 2020, Göhring, Kellner, Schrangl, Platzer, Klotzsch, Stockinger, & Schütz, for example, found that FRET-based sensor equipped with a TCR-reactive single-chain antibody fragment, which was tethered to planar supported lipid bilayers (SLBs) and informs most directly on the magnitude and kinetics of TCR-imposed forces at the single-molecule level. Thus, focusing on "a systematic approach coupling FRET and sequencing (FRET-Seq) to integrate random mutagenesis, fluorescence-activated cell sorting (FACS), and next-generation sequencing (NGS) to screen and identify sensitive biosensors from large-scale libraries directly in mammalian cells, utilizing the design of self-activating FRET (saFRET) biosensor" seems challenging and important.

However, the research's basic strategy was to combine "several techniques to systematically and efficiently develop FRET biosensors". Yet, Fyn and ZAP70 kinase biosensors were chosen as the targets in this study, and while carefully demonstrating that "HTDS assay revealed that Staurosporine, AZD7762, and Tie2 kinase inhibitor from the compound library can effectively inhibit the ZAP70 activity" using "a disease model where T cell activation is mediated by ZAP70-R360P mutation" seems to reduce the scope of benefits of the model. In other words, declaring that this study has established a platform to screen large-scale biosensor libraries in mammalian cells for cellular imaging and drug screening", seems a little excessive. I thus concluded that in the absence of the demonstration of broader biological relevance the paper is more suitable for a specialized journal.

In addition, as the authors noted in line 81, the FRET-HTDS assays were successfully used for monitoring insulin-receptor activation, SERCA2a-PLB interaction, and PKA kinase activity in previous studies. This study thus suffers from lack of originality and novelty.

Additional comments

- In the stage of calculating the ER value, normalization to the WT in the same library will allow us to compare the affinity of each variant in a manner that is not influenced by the size of the libraries or the frequency of the mutant in the original library. Here, normalization was performed only according to the frequency of all variants without reference to WT. Calibration according to WT frequencies is thus required.
- In Figure 2d: Dynamic changes(%)- the error bar of WT is large. Is the difference statistically significant? A statistical test is needed to conclude (in line 150) that multiple biosensor variants were identified to have significantly improved dynamic changes comparing to the parent Fyn biosensor upon treatment with PP1.
- Figure 2e: This figure is very messy and unclear. If one wants to look at a specific variant, it is almost impossible.
- Line 122: In the study, they created 2 two biosensor libraries (Lib1: -1, -2, -3, Y or Lib2: Y, +1, +2, +3) with each library consisting of 32,768 variants. (but the variety of $19 \times 3 = 6,859$ different variants is possible in each library). Therefore, there is no clarity regarding the variety of 32,768 variants reported in the article. This clarity is critical since all variants are taken into account while calculating the frequency and value of the ER.
- Line 166: The article stated that the combination of two improved mutants from Lib1 and Lib2 did not improve the performance of the biosensor. In the Supplementary (Fig. 4) the authors show 3 combined variants, and on this basis, they determine that there is an uncooperative effect of

amino acids up-and down-stream of tyrosine in the substrate. Is that enough to establish this claim? No test is performed on a comprehensive number of variants (more than 3 variants). Therefore, data on more variants should be added.

- Line 124: After the generation of a mammalian cell library by infection with viral libraries, individual cells expressing biosensor variants were sorted. No details were given in the paper on the infection process of the viruses (and on the MOI value). How for example the authors ensured that indeed each cell was infected with only one variant?

Reviewer #2:

Remarks to the Author:

The manuscript by Liu and colleagues describes the generation of new FRET-based biosensors for Fyn and Zap70 kinases. To identify sensitive biosensors the authors present a new methodology based on biosensor libraries expressed and screened in mammalian cells via FACS. Three residues in the biosensor interface around a consensus tyrosine residue were randomized, respectively, and biosensors screened for sensitivity. Retrieved variants were characterized in T-lymphocytes. Overall, it is acknowledged that screening in mammalian cells could possibly yield more sensitive FRET sensors. The kinase-on or kinase-dead variants of the sensors offer the advantage that sensors can be screened under signal-on or signal-off conditions. Overall, however, it seems that the method is insufficiently developed. Only few sites around the phosphorylated tyrosine were randomized. This approach may surely contribute to increasing sensitivity. Why weren't linkers between FPs and the other sensor domains randomized? This should have a large effect on overall FRET and sensitivity. Sensor variants generated still have a modest signal strength compared to other FRET sensors that had been made using conventional engineering. Thus, the methodology proposed here is not convincing at this stage.

Minor:

-Some of the graphs need to be improved, e.g 2b, 2e, 3d. Hard to recognize detail in view of the many dots and lines.

Fig 1d: Since PP1 inhibitor is reversible, what is the reason for the stabilization of the FRET ratio to a plateau? Is ECFP – Ypet interaction preventing phosphorylation of the substrate by newly operative kinase domain? While this is profitable for HTDS screening of kinase inhibitors, it compromises the use of the sensor in imaging the cellular dynamic of tyrosine kinase signaling.

1d: color code for SH2 domain, an schematic representation of linker regions and substrate domain, should be kept consistent to not confuse the reader.

Fig- 1,3,5: y-axis labelling: consistency

Fig. 3 f: Disturbing high FRET ratio dot on represented cell for the variant SREYYDM. What is this?

Reviewer #3:

Remarks to the Author:

In this manuscript by Lui et al, the authors describe a novel high throughput method to produce biosensors for tyrosine kinases. This method does produce more high efficiency biosensors that would have usefulness to the field. However, there are several changes needed to greatly strengthen the conclusions of this paper.

1) We appreciate the amount of effort in Figure 2D and 2E but the addition of all the data in Figure 2E makes it difficult to interpret. Could the authors only show a few key mutations for Figure 2E that would better highlight classes of mutations. Also, could the authors make the color coding identical from Figure 2D and 2E?

2) The authors describe several experiments in Figure 4 that show altered localization of the ZAP-70 sensor after stimulation with antibody clusters or target cells. These experiments provide evidence for the in vivo usefulness of this sensor to provide spatial and temporal localization of ZAP-70 after TCR or CAR stimulation. However, the images that are obtained are of so low resolution that it is impossible to identify ZAP-70 sensor localization after stimulation. Have the

authors thought of using higher resolution microscopy techniques to more precisely localize the activation of ZAP-70 or Fyn? The addition of higher resolution methods with analysis would benefit our understanding of Fyn and/or ZAP-70 biology.

3) The screen shown in Figure 5 found several inhibitors that reduced ZAP-70 kinase function. They then tested two of these compounds, staurosporine and AZD7762, in in vivo assays using Jurkat T cells. These compounds were able to reduce TCR-induced phosphorylation of ZAP-70 and LAT in cells expressing WT ZAP-70 and R360P ZAP-70. The authors then conclude that “the inhibitors identified through the saFRET-HTDS assay can efficiently inhibit the ZAP70 kinase signaling pathway” and state that these inhibitors “have therapeutic potentials in treating diseases involving abnormal ZAP70 kinase or T cell activation.” They also state in the abstract that “the improved saFRET biosensors further allowed the identification of compounds that demonstrated novel efficiency in inhibiting ZAP70 kinase activity and disease-related T-cell activation.” Staurosporine and AZD7762 certainly inhibit ZAP-70, but they also broadly inhibit many other kinases that are relevant for TCR signaling pathways. In fact, staurosporine is well known as a general tyrosine kinase inhibitor. The reduced phosphorylation of ZAP-70 and LAT by staurosporine and AZD7762 could be due to their inhibition of LCK, FYN and several other kinases. In addition, these drugs do not have therapeutic potential. Staurosporine is highly toxic to multiple cells and AZD7763 was withdrawn from clinical trials due to cardiotoxic effects. The authors show the toxic effects of these drugs in Figure 6F and 6L, where there is a large change in side scatter after treatment, and in Supplemental Video 7, where the drugs cause a marked change in cell morphology. In the end, the screen is novel and can identify inhibitors of ZAP-70 but the inhibitors identified in the described screen also inhibit many other kinases. Figure 6 has little relevance other than to confirm that staurosporine and AZD7762 are good kinase inhibitors and the statements describing this work are not supported by the data.

Response to the reviewers

We sincerely thank the editor and the reviewers for your time and effort, with the constructive feedbacks. Following the suggestions from the reviewers, we have conducted substantial experiments and revised our manuscript thoroughly. Specifically, we have applied the FRET biosensor optimized by FRET-seq to (1) visualize the ZAP70 activation patterns at subcellular levels with high resolutions in T cells upon TCR or CAR engagement, and (2) elucidate the role of ITAM in regulating CAR-T cell functions by quantifying and comparing the differential levels of ZAP70 activation in T cells expressing CARs with varying ITAM motifs in the cytoplasmic tails. We also demonstrated the superiority of the saFRET assay in minimizing the false positive rate of selecting non-specific inhibitors of ZAP70 over the conventional FRET biosensor screening assay. Employing the saFRET drug screening assay based on the optimized ZAP70 biosensor, we are excited to discover that an FDA-approved drug Sunitinib, which was originally approved to treat renal cell carcinoma (RCC) and imatinib-resistant gastrointestinal stromal tumor (GIST), can be repurposed to inhibit the ZAP70 kinase and its hyperactive ZAP70 R360P mutant, which is the main cause of an autoimmune disease requiring allogeneic hematopoietic cell transplantation (HCT) in patients¹. As such, we believe that this revision has led to a significant improvement of our manuscript, for which we want to thank the reviewers. Here, we provide a point-by-point response to the comments from all reviewers, which is also incorporated into our revised manuscript.

REVIEWER COMMENTS

Reviewer #1 (Remarks to the Author):

REVIEW

In general:

The attempt to establish a platform to screen large-scale biosensor libraries in mammalian cells for cellular imaging and drug screening is not a simple matter and it has called for the attention of academic researchers. In 2020, Göhring, Kellner, Schrangl, Platzer, Klotzsch, Stockinger, & Schütz, for example, found that FRET-based sensor equipped with a TCR-reactive single-chain antibody fragment, which was tethered to planar supported lipid bilayers (SLBs) and informs most directly on the magnitude and kinetics of TCR-imposed forces at the single-molecule level. Thus, focusing on "a systematic approach coupling FRET and sequencing (FRET-Seq) to integrate random mutagenesis, fluorescence-activated cell sorting (FACS), and next-generation sequencing (NGS) to screen and identify sensitive biosensors from large-scale libraries directly in mammalian cells, utilizing the design of self-activating FRET (saFRET) biosensor" seems challenging and important.

However, the research's basic strategy was to combine "several techniques to systematically and efficiently develop FRET biosensors". Yet, Fyn and ZAP70 kinase biosensors were chosen as the targets in this study, and while carefully demonstrating that "HTDS assay revealed that Staurosporine, AZD7762, and Tie2 kinase inhibitor from the compound library can effectively inhibit the ZAP70 activity" using "a disease model where T cell activation is mediated by ZAP70-R360P mutation" seems to reduce the scope of benefits of the model. In other words, declaring that this study has established a platform to screen large-scale biosensor libraries in mammalian cells for cellular imaging and drug screening", seems a little excessive. I thus concluded

that in the absence of the demonstration of broader biological relevance the paper is more suitable for a specialized journal.

Response: We thank the reviewer for the encouraging comments and the recognition of the importance of our work. We believe that this is the first time that FRET screening and NGS are integrated to optimize biosensors from large-scale libraries directly in mammalian cells. Biosensors for the two different kinases Fyn and ZAP70 were chosen as the representatives to demonstrate the feasibility of our platform for broad applications as our design is modular and can be readily extended for optimizing different kinase biosensors. Similarly, we used the ZAP70 saFRET biosensor as one representative example to showcase the applicability of our drug screening platform for different kinase inhibitors. We agree with the reviewer that FRET biosensors can be applied to quantify the magnitude and kinetics of TCR-imposed forces, and hence providing powerful tools to reveal molecular insights underlying T cell physiology. Accordingly, we have cited the excellent paper that the reviewer mentioned.

We also agree with the reviewer that the demonstration of broader biological relevance of our work would be helpful to highlight the power of our platform. In fact, despite the crucial roles of ZAP70 kinase in T-cell functions, the tracking of ZAP70 kinase activity using the previously available biosensors was very limited, mainly due to the rather poor sensitivity of the biosensors. The relatively weak intrinsic activities of ZAP70 kinase further added the detection difficulty². For example, the first ZAP70 biosensor (ROZA developed in 2008 by Annemarie Coffman Lellouch's group) showed only ~5-8% dynamic change after the physiological anti-CD3 stimulation in Jurkat T cells (see *figure 2B of the cited paper, top panel*)³. The second version of the ZAP70 FRET biosensor developed in 2015 by the same group also had highly heterogeneous responses and significant overlap in signals between unstimulated and stimulated cell populations (see *figure 3A-B of the cited paper*)⁴. In contrast, our new biosensor optimized by FRET-seq had ~25% and ~40% changes in Jurkat and primary T cells with high specificity upon anti-CD3/CD28 stimulation, respectively (Fig. 4c, supplementary Fig. 9f). The optimization of saFRET biosensors also allowed the high throughput screening of inhibitors which was not possible with an older version of the biosensor (Supplementary Fig. 6b). These results indicate that the biosensors newly developed using our FRET-seq platform can allow new biological studies and applications.

Following the reviewer's suggestion, we have further applied the ZAP70 biosensor to investigate the underlying mechanism of different CAR (chimeric antigen receptor) functions in T cells to demonstrate the broader biological relevance of our work. T Cell-based immunotherapy, e.g., CAR-T therapy, has revolutionized cancer treatment. The second-generation design of CARs, containing a CD28 or 4-1BB-derived costimulatory domain at the cytoplasmic tail of CAR, has been widely applied in the clinic⁵. Different designs of these CAR molecules, with varying immunoreceptor tyrosine-based activation motifs (ITAMs) at the CAR cytoplasmic tail, have been shown to result in different anti-tumor potencies *in vivo*⁶. However, the mechanism of ITAM in regulating CAR T cell functions, which is crucial for the design of new CARs, remains unclear. Since both ZAP70 and ERK kinases can be regulated by the CAR cytoplasmic tail and serve as key effectors for CAR signaling and T cell activation^{7,8}, we examined the role of ITAM motif in regulating the ZAP70 and ERK kinases in response to CAR activation, utilizing our optimized ZAP70 biosensor and an ERK biosensor with high sensitivity⁹. These FRET

biosensors were co-expressed with the wild type CAR (1928 ζ , WT-CAR) or its mutated version (1928 ζ , XX3-CAR), which had inferior anti-tumor efficacy than its wild-type counterpart (the tyrosine sites of the first two ITAM motifs in the CAR cytoplasmic tail have been mutated to phenylalanines) ⁶. The kinase activity was tracked by live-cell imaging after stimulating the CAR-T cells expressing either WT-CAR or XX3-CAR with antigen-presenting CD19⁺ 3T3 cells (New Fig. 5a). Large FRET ratio changes of ERK and ZAP70 biosensors in both types of CAR-T cells were observed when engaging with the CD19⁺ 3T3 cells, indicating that both ZAP70 and ERK kinases are responsive to antigen stimulation (New Fig. 5b-g). The ERK signals increased within 5 min after antigen engagement and quickly reached a plateau (New Fig. 5b-c). However, no significant difference in ERK activity was observed between CAR-T cells expressing WT-CAR or XX3-CAR (New Fig. 5d). In contrast, a significantly delayed response and a reduced activation level of ZAP70 kinase activity were observed in XX3-CAR T cells comparing to WT-CAR T cells (New Fig. 5b-g), suggesting a significant ZAP70 defect in XX3-CAR-T cells. To verify the results of single cell imaging, we further evaluated ZAP70 activity in cells expressing different CARs in a large-scale, high-throughput manner using flow cytometer. Consistent with these findings, a remarkable increase in the percentage of the ZAP70-active CAR T cells (as indicated by high-FRET ratios) was observed after CD19⁺ Raji stimulation (New Fig. 5h-i). There was a significantly higher proportion of ZAP70-active cells in WT-CAR T cells than that of XX3-CAR T cells (New Fig. 5j-l), confirming that ZAP70 was more activated by WT-CAR than by XX3-CAR.

These results indicate that ITAM motifs may affect ZAP70, but ERK, to modulate CAR T cell functions. Our results also suggest that our optimized ZAP70 biosensor is applicable to monitor and evaluate the signaling of different CAR molecules and enable the high-throughput screening and selection of functional CARs by FACS. We believe that these new findings have provided deeper insights into the role of ZAP70 kinase in different CARs-signaling regulations, highlighting the power of our new biosensor in applications with broader biological relevance. We want to thank the reviewer for stimulating these new experiments/results, which led to a significant improvement of our manuscript. We have revised the manuscript accordingly (Introduction, P3; Results, P10-11; Figure 5).

New Figure 5. Less ZAP70 activation in XX3-CD19 CAR-T cells.

a, Schematic drawings of constructs. The ERK- or ZAP70-FRET biosensor was co-expressed with WT-CAR or XX3-CAR in Jurkat T cells. In XX3-CAR, the tyrosine in the first and second ITAM motif was mutated to phenylalanine. The CAR T cells were then dropped onto the 3T3 cells that constitutively express CD19 to monitor the dynamic ZAP70 or ERK kinase activations.

b, Representative images of ERK biosensor in T cells after attaching to the 3T3 CD19+ cell monolayer. Scale bar=10 μ m.

c, Time courses of FRET ratio (FRET/CFP) of ERK biosensor in WT CAR-T or XX3-CAR-T cells (N=8 and 20 in each group). Error bars, mean \pm SEM.

d, Percentage changes of ERK-FRET biosensor in WT CAR-T or XX3-CAR-T cells (Unpaired two-tailed Student's t-test, NS, $P>0.05$). Error bars, mean \pm SD.

e, Representative images of ZAP70 biosensor in T cells after attaching to the 3T3 CD19+ cell monolayer. Scale bar=10 μ m.

f, Time courses of FRET ratio (ECFP/FRET) of ZAP70 biosensor in WT CAR-T or XX3-CAR-T cells (N=13 and 10 in each group). Error bars, mean \pm SEM.

g, Percentage changes of ZAP70-FRET biosensor in WT CAR-T or XX3-CAR-T cells (Unpaired two-tailed Student's t-test, *P=0.017). Error bars, mean \pm SD.

New Figure 5. Less ZAP70 activation in XX3-CD19 CAR-T cells.

h, Flow-cytometry analysis of ECFP/FRET in WT-CAR T cells before and after CD19⁺ Raji cell stimulation.

i, Flow-cytometry analysis of ECFP/FRET in XX3-CAR T cells before and after CD19⁺ Raji cell stimulation.

j, Percentage of High-FRET ratio cells in different groups of three independent experiments. (One-way ANOVA, ****p<0.0001, **P=0.0054). Error bars, mean \pm SD.

k. Normalized FRET ratio of WT- or XX3 CAR-T cells before and after CD19⁺ Raji cell stimulation. (One-way ANOVA, ****P<0.0001). Error bars, mean \pm SEM.

l, Histogram of FRET ratio in WT- or XX3 CAR-T cells after CD19⁺ Raji cell stimulation.

In addition, as the authors noted in line 81, the FRET-HTDS assays were successfully used for monitoring insulin-receptor activation, SERCA2a-PLB interaction, and PKA kinase activity in previous studies. This study thus suffers from lack of originality and novelty.

Response: Our saFRET-HTDS assay is novel and fundamentally different from the conventional FRET assays. First of all, since saFRET biosensor has the corresponding kinase domain fused to the FRET biosensor, this design enables us to screen small molecule inhibitors in adherent cells for kinases typically expressed in suspension cells, e.g. ZAP70, thus overcoming the limitation and difficulty of imaging in suspension cells. Furthermore, compared to conventional assays, saFRET-HTDS enables the screening of inhibitors directly targeting ZAP70 kinase with high specificity, avoiding the false positive selection of inhibitors targeting upstream molecules instead of ZAP70 itself in host cells. Indeed, our results showed that the hits using the conventional FRET biosensor assay include inhibitors that target ZAP70 upstream signaling molecules, but not ZAP70 itself (New Supplementary Fig. 10e-f). These false-positive hits were successfully eliminated in our saFRET-HTDS assay (New Fig. 6c and New supplementary Fig. 10g-h). These results demonstrate the higher specificity of our saFRET screening assays over the conventional FRET assays.

In summary, saFRET biosensor design provides a platform for biosensor-based high-throughput drug screening (HTDS) in living cells. The carry-on-kinase-domain should bypass the need of maintaining endogenous regulation of the target kinase, minimize the heterogeneity and noise of endogenous kinase activation, avoid identifying pathway regulators instead of on-target hits, and provide flexibility of choosing suitable cell systems that maintain live-cell contexts but offer experimental ease. In the future, the counter-screening strategy incorporating a kinase-active or kinase-dead domain in biosensor variants can further promote the specificity of inhibitors identified in our saFRET assays. We have added these new results and discussions into the revised manuscript (Results P11 and P12; discussion P14).

New Sup Figure 10. The saFRET assay is specific and immune to upstream signaling of the target kinase.

e, Schematics of conventional FRET assay in T cells. The FRET change could be affected by inhibitors targeting ZAP70 kinase and its upstream molecules.

f, The Src, Fyn, Lck, and Syk kinase inhibitors could reduce the FRET change of conventional biosensors after CD3/CD28 antibody stimulation significantly. The inhibitor groups are pretreated with corresponding inhibitors as indicated for 30 min before antibody stimulation. Control indicates the DMSO pre-treated group. (n>12 in each group, One-way ANOVA, ****P<0.0001 and **P=0.0013).

g, Schematics of saFRET assay in HEK293 cells. The FRET change is mediated by the kinase domain.

h, Inhibitors of Src, Fyn, Lck, and Syk kinases could not cause a significant FRET change of the ZAP70 saFRET biosensor. Control indicates the DMSO treated group. (n>15 in each group and data were tested by One-way ANOVA).

Additional comments

• In the stage of calculating the ER value, normalization to the WT in the same library will allow us to compare the affinity of each variant in a manner that is not influenced by the size of the libraries or the frequency of the mutant in the original library. Here, normalization was performed only according to the frequency of all variants without reference to WT. Calibration according to WT frequencies is thus required.

Response: We have followed the reviewer’s suggestion and calibrated the ER values of variants to that of WT. Indeed, the probability of identifying improved biosensors increased when selecting the variants that have a larger ER value than WT: the

positive rate increased from 60% to 78.2% after further calibration using the WT biosensor (New Supplementary 3d and supplementary table 2). We thank the reviewer for this excellent suggestion, which led to the improvement of our manuscript. We have added these new results into the revised manuscript (Results P7).

New Supplementary Figure 3d. Calibration using the WT biosensor as a reference improved the optimization success rate.

d. The relation between the experimentally verified dynamic range (%) of biosensor variants and their product of $E_v(\text{KAH})$ and $E_v(\text{KDL})$. The dash lines represent the measured dynamic change (across y-axis) and the value of $E_v(\text{KAH}) \times E_v(\text{KDL})$ (across x-axis) of wild-type biosensor. Red dots, blue dots, and the red star represent the better biosensor, worse biosensor, and WT biosensor, respectively. 37% and 78% represent the positive rates of identifying a better biosensor when the product values on X-axis are below or above that of the WT biosensor, respectively. 60% represents the overall positive rate without the WT biosensor as a reference.

- In Figure 2d: Dynamic changes (%)- the error bar of WT is large. Is the difference statistically significant? A statistical test is needed to conclude (in line 150) that multiple biosensor variants were identified to have significantly improved dynamic changes comparing to the parent Fyn biosensor upon treatment with PP1.

Response: We thank the reviewer for this suggestion. Yes, there are multiple variants having statistically significant difference from the WT biosensor. We have added the statistical test as the reviewer suggested in Figure 2d, as well as in Figure 3g, Supplementary Figure 3c.

- Figure 2e: This figure is very messy and unclear. If one wants to look at a specific variant, it is almost impossible.

Response: We have revised this figure and only present the 5 candidates in the main figure 2e to streamline the presentation. We also simplified the Figure 2d, and Figure 2f accordingly.

- Line 122: In the study, they created 2 two biosensor libraries (Lib1: -1, -2, -3, Y or Lib2: Y, +1, +2, +3) with each library consisting of 32,768 variants. (but the variety of $19 \times 3 = 6,859$ different variants is possible in each library). Therefore, there is no

clarity regarding the variety of 32,768 variants reported in the article. This clarity is critical since all variants are taken into account while calculating the frequency and value of the ER.

Response: In this study, 32768 ($32 \times 32 \times 32$) is the diversity of nucleotide sequences which corresponds to 8000 amino acid sequences ($20 \times 20 \times 20$). We have clarified this in the revised manuscript (Results, P6 and Supplementary methods and material file, P2).

- Line 166: The article stated that the combination of two improved mutants from Lib1 and Lib2 did not improve the performance of the biosensor. In the Supplementary (Fig. 4) the authors show 3 combined variants, and on this basis, they determine that there is an uncooperative effect of amino acids up-and down-stream of tyrosine in the substrate. Is that enough to establish this claim? No test is performed on a comprehensive number of variants (more than 3 variants). Therefore, data on more variants should be added.

Response: We thank the reviewer for this suggestion. We conducted additional experiments and have tested another five combinations (Dash line box in New Supplementary Fig. 5a represents the new combinations) of selected mutants from Lib1 and Lib2, using live-cell imaging as the reviewer suggested. Similar to the previous observations, direct combination of the preferred substrate sequences from these two separate libraries did not lead to further improved biosensors. We have added these new results into the supplementary figure 5 in the revised manuscript.

New Supplementary Figure 5. The combinations of improved mutants from Lib 1 and Lib2.

a. Comparison of the biosensors with combined substrate sequences from both Lib 1 and Lib 2 vs their parental biosensors from either Lib 1 or Lib2. Star indicates the biosensors with combined substrate sequences (left columns). The middle columns are biosensors from Lib1 and the right columns from Lib2. The dashed lines indicate the mean FRET change of original WT (EGTYGVV) biosensor. ($N \geq 15$ for each group, One-way ANOVA, **** $P < 0.0001$, ** $P = 0.0038$, NS=Not significant).

b. Time courses of the ECFP/FRET ratio signals of the combined biosensors after PP1 treatment. Error bars, Mean \pm SD.

• Line 124: After the generation of a mammalian cell library by infection with viral libraries, individual cells expressing biosensor variants were sorted. No details were given in the paper on the infection process of the viruses (and on the MOI value). How for example the authors ensured that indeed each cell was infected with only one variant?

Response: The value of MOI is 0.1, which is specified in the revised manuscript and supplementary methods. ~95% of the infected cells in our experiments have only one variant, which can be assessed by the following analysis: the ratios of functional viruses to target cells can be described in terms of Poisson distributions¹⁰; thus, the percentage of the cells not infected can be calculated by $P(0) = e^{-0.1} = 90.48\%$, and the percentage of cells infected with only one virus is $P(1) = 0.1e^{-0.1} = 9.05\%$; the remaining 0.47% cells have more than one variant (See also: <https://www.virology.ws/2011/01/13/multiplicity-of-infection>); we eliminated the non-infected cells via FACS; as a result, ~95% of the infected cells have only one variant in the sorted cells which were subjected to further experiments. The enrichment ratio was used to rank the variants and counter-sorting was further used to eliminate the noises engendered during screening and sequencing. We have added the above details into the revised supplementary method file (Supplementary methods and material file, P3-4).

Reviewer #2 (Remarks to the Author):

The manuscript by Liu and colleagues describes the generation of new FRET-based biosensors for Fyn and Zap70 kinases. To identify sensitive biosensors the authors present a new methodology based on biosensor libraries expressed and screened in mammalian cells via FACS. Three residues in the biosensor interface around a consensus tyrosine residue were randomized, respectively, and biosensors screened for sensitivity. Retrieved variants were characterized in T-lymphocytes.

Overall, it is acknowledged that screening in mammalian cells could possibly yield more sensitive FRET sensors. The kinase-on or kinase-dead variants of the sensors offer the advantage that sensors can be screened under signal-on or signal-off conditions.

Overall, however, it seems that the method is insufficiently developed. Only few sites around the phosphorylated tyrosine were randomized. This approach may surely contribute to increasing sensitivity. Why weren't linkers between FPs and the other sensor domains randomized? This should have a large effect on overall FRET and sensitivity.

Response: We thank the reviewer for the encouraging notes and suggestions. For the biosensor optimization, it was reported that a few sites in the biosensor can change the biosensor performance significantly. For example, our previous study found a single mutation in the SH2 domain (C185A) together with a mutation in the substrate could significantly improve the performance of the SRC kinase FRET biosensor ¹¹. Furthermore, our previous work (from J. Zhang's lab) showed that optimizing only four amino acids in the linker can improve the single-FP biosensor significantly ¹². Thus, for the FRET biosensor optimization, the change of a few amino acids can be sufficient to improve the performance of the biosensors.

Previous studies suggest that amino acids surrounding the tyrosine residue are sufficient to affect the preference of substrate to protein kinase and SH2 domain. In fact, previous studies have proven that three amino acids surrounding the tyrosine (Y) residue (-1, -2, -3, Y, +1, +2, +3) on the substrate peptide are the most critical for kinase phosphorylation and SH2 interaction ^{13, 14}. Thus, we started the substrate optimization with three amino acids neighboring the consensus tyrosine site. Since the substrate peptide of the FRET biosensor tested in this study is in the central part of the biosensor connecting to the linker region, we expect that changing substrate sequence should affect the linker function that influence the FPs orientation.

Taken all together, while there are only a few sites for random mutagenesis, the diversity of the library and the coverage should be appropriate for screening the biosensor library in mammalian cells. Moreover, as the reviewer also noted, this FRETseq platform with its modular design can be readily extended to directly optimize the linker or SH2 domain in future studies.

To verify whether the increase of the mutation residues in the substrate could further enhance the performance of the biosensor, we performed a 2nd round optimization. An improved Fyn biosensor from the first-round screening (DYGYGVV) was further optimized by mutating additional three amino acids after the tyrosine residue (Y, +1, +2, +3) (Figure 1 for Reviewers). Our results indicate that the improvement of the biosensor sensitivity is rather mild compared to the 1st round screening. These results indicate that one-round optimization of a few residues in the

substrate should be sufficient for tyrosine kinase FRET biosensor optimization in our FRET-seq platform.

Figure 1 for Reviewers: The 2nd round screening of Fyn FRET biosensor based on DYGYGVV identified in Lib1.

a, The 4D plot of the enrichment ratio of substrates in different groups for 2nd round library (DYGXXXX), in which the amino acid residues after the consensus tyrosine were mutated. The enrichment ratio of the biosensors in the KAH group was color-coded. The substrates satisfying all four criteria were highlighted in color.

b, Scatter plot of the enrichment ratio of biosensor variants. Red dots represent the top 10 biosensor variants that we tested.

c, Quantification of the FRET dynamic change (%) of selected biosensor variants upon PP1 treatment ($n \geq 15$ in each group; one biosensor is not significant compared to WT biosensor as indicated; NS: not significant). Error bars, mean \pm SEM.

d, Quantification of the normalized dynamic ECFP/FRET ratio of the representative improved biosensors that have been tested. FRET ratio change of the parental biosensor was marked in gray dash line ($n \geq 15$ in each group). Error bars, Mean \pm SEM.

Sensor variants generated still have a modest signal strength compared to other FRET sensors that had been made using conventional engineering. Thus, the methodology proposed here is not convincing at this stage.

Response: ZAP70 kinase is crucial for T-cell functions and a broad range of inflammatory/autoimmune diseases, but its relatively weak intrinsic activity led to the detection difficulty². The previously available biosensors for ZAP70 kinase suffered from low specificity and dynamic range. Indeed, (1) the first ZAP70 biosensor (ROZA developed in 2008 by Annemarie Coffman Lellouch's group)³ showed **~only 5-8% dynamic change** after the physiological anti-CD3 stimulate in Jurkat T cells (figure 2B, top for Jurkat)³; (2) The 2nd generation of the ZAP70 FRET biosensor (ROZAXL) developed in 2015 by the same group⁴ also had highly heterogeneous responses and significant overlap in signals between unstimulated vs. stimulated cell populations (see Figure 3 A-B in ref⁴); (3) An SYK kinase FRET biosensor was developed by us for monitoring SYK kinase activity in T cells, with a peptide sequence from SYK substrate VAV2, which was later found to respond non-specifically to ZAP70 kinase activity^{15,16}. We have tested the response of all these three substrates in our saFRET biosensor framework (using the improved SH2 domain with C185A mutation)¹¹ to have a fair comparison. **A non-specific response was observed** in ZAP70 kinase-dead biosensors with substrates from ROZA and SYK biosensor (supplementary figure 6). The biosensor using **the substrate sequence from ROZAXL had a ~6% of FRET ratio changes** after anti-CD3 stimulation in Jurkat T cells, with non-specific responses in ZAP70 deficient P116 cells upon TCR activation (Figure 4e).

In contrast, our new biosensor optimized by FRET-seq had ~25% and ~40% changes, respectively, in Jurkat and primary T cells with high specificity upon anti-CD3/CD28 stimulation (Fig. 4c, supplementary Fig. 9f). Thus, FRET-seq platform can be used to improve the biosensor performance significantly, particularly for the difficult-to-detect ZAP70 kinase activity. We further applied our new ZAP70 FRET biosensor to study ZAP70 activation in T cells. TCR engagement clearly caused a ZAP70 activation with organized distribution at subcellular levels, concentrating at the central TCR clusters and the cell periphery where cortical actin bundles are localized. This spatially organized distribution of ZAP70 activity in response to TCR engagement is significantly different from that of CAR stimulation (Fig. 4f-j and New supplementary Fig. 9g-h). The optimized ZAP70 biosensor further allowed us to examine the underlying regulatory mechanism of different CAR designs in T cells. Our results showed that wild-type CAR (2nd generation) and its cytoplasmic tail mutated version XX3 (the tyrosine sites of first two ITAM motifs have been mutated to phenylalanines) had clear difference in detecting antigen-presenting cells, as monitored by the optimized ZAP70 biosensor (New Fig. 5). As such, we believe that these newly optimized biosensors allowed a deeper understanding of the activation pattern of ZAP70 kinase and its role in CAR- and TCR-signaling regulations, highlighting the power of these new biosensors for broad biological applications. We have added these new results and the above discussions into the revised manuscript in the discussion part (Introduction, P3; Results, P10-11; Figure 5).

Minor:

-Some of the graphs need to be improved, e.g 2b, 2e, 3d. Hard to recognize detail in view of the many dots and lines.

Response: We have followed the reviewer’s suggestion and simplified the figures accordingly. Specifically, we have revised figure 2e to streamline the presentation by only presenting 5 biosensors. We also simplified Figure 2d and Figure 3h, and provided high-resolution files in this revision to show the enrichment ratio of all variants.

Fig 1d: Since PP1 inhibitor is reversible, what is the reason for the stabilization of the FRET ratio to a plateau? Is ECFP – Ypet interaction preventing phosphorylation of the substrate by newly operative kinase domain?

Response: During the biosensor testing using live-cell imaging, PP1 was kept in the medium, thus keeping the FRET ratio at a low level. Following the reviewer’s suggestion, we tested the reversibility of the saFRET biosensor by wash out PP1 during imaging. The saFRET biosensor was also reversible as washing out the PP1 during imaging led to the recovery of the FRET ratio to the original value, in contrast to the control no-wash group (New Supplementary Fig. 1). We have added the above results into the supplementary figure 1 in this revised manuscript.

New supplementary Figure 1. The saFRET biosensor is reversible.

a-b, Representative images (a) and time courses (b) of the FRET (ECFP/FRET) ratio signals of the Fyn saFRET biosensors (Substrate: DYGYGVV) in different groups. PP1 was washed out during imaging in the wash-out group, while kept in the medium

in the no-wash group (N=14 in each group). Error bars, mean \pm SEM. Scale bars, 10 μ m.

While this is profitable for HTDS screening of kinase inhibitors, it compromises the use of the sensor in imaging the cellular dynamic of tyrosine kinase signaling.

Response: The saFRET was designed to facilitate the biosensor optimization and drug screening, with the kinase domain fused to the biosensor to eliminate the influence of heterogeneously expressed endogenous kinases in host cells. For live-cell imaging, the kinase domain in the biosensor will be removed to monitor endogenous tyrosine kinase activities in cells.

1d: color code for SH2 domain, an schematic representation of linker regions and substrate domain, should be kept consistent to not confuse the reader.

Response: We thank the reviewer for this suggestion. We have changed the color code for SH2 domain accordingly.

Fig- 1,3,5: y-axis labelling: consistency

Response: We have changed the figures to make the labeling consistent across all the Figures, including the figures in the supplementary materials.

Fig. 3 f: Disturbing high FRET ratio dot on represented cell for the variant SREYYDM. What is this?

Response: We have checked all the cells tested in this group and found that this phenomenon is rare. In all the 21 imaging positions examined, only 2 cells showed this disturbing high FRET ratio dot which did not change during imaging. This phenomenon is possibly due to the abnormal aggregation of the biosensor in cell derbies attached to the cell surface. We have checked the phase and fluorescence images of similar cells to demonstrate the abnormal aggregate in the cells.

Figure 2 for Reviewers: DIC and FRET ratio image of an abnormal high-FRET ratio dot in cells

Reviewer #3 (Remarks to the Author):

In this manuscript by Lui et al, the authors describe a novel high throughput method to produce biosensors for tyrosine kinases. This method does produce more high efficiency biosensors that would have usefulness to the field. However, there are several changes needed to greatly strengthen the conclusions of this paper.

Response: We thank the reviewer for the efforts in evaluating our manuscript and providing suggestions to improve our manuscript, especially for the application of the biosensors in T cell imaging and the drug screening assay using the newly developed saFRET biosensor. While the main focus and novelty is the development of the FRET-seq platform, we have shown that the optimized saFRET biosensor enables the screening of ZAP70 inhibitor in a high-throughput manner. In fact, employing the saFRET drug screening assay based on the optimized ZAP70 biosensor, we further discovered that an FDA-approved drug Sunitinib, which was originally approved to treat renal cell carcinoma (RCC) and imatinib-resistant gastrointestinal stromal tumor (GIST), can be repurposed to inhibit the ZAP70 kinase and its hyperactive ZAP70 R360P mutant, which is the main cause of an autoimmune disease requiring allogeneic hematopoietic cell transplantation (HCT) in patients¹. We have performed additional experiments to show that the saFRET assay has higher specificity than the conventional FRET assay in screening ZAP70 inhibitors. We also applied our optimized biosensor in high-resolution imaging of immunological synapse of T cells and CAT-T cells, in which the spatial difference of TCR and CAR molecules was observed.

1) We appreciate the amount of effort in Figure 2D and 2E but the addition of all the data in Figure 2E makes it difficult to interpret. Could the authors only show a few key mutations for Figure 2E that would better highlight classes of mutations? Also, could the authors make the color coding identical from Figure 2D and 2E?

Response: We thank the reviewer for this suggestion. We have revised figure 2d-e and streamlined the presentation by only presenting 5 biosensors in figure 2e. We also changed the color code of the variants to make them consistent across different figures.

2) The authors describe several experiments in Figure 4 that show altered localization of the ZAP-70 sensor after stimulation with antibody clusters or target cells. These experiments provide evidence for the in vivo usefulness of this sensor to provide spatial and temporal localization of ZAP-70 after TCR or CAR stimulation. However, the images that are obtained are of so low resolution that it is impossible to identify ZAP-70 sensor localization after stimulation. Have the authors thought of using higher resolution microscopy techniques to more precisely localize the activation of ZAP-70 or Fyn? The addition of higher resolution methods with analysis would benefit our understanding of Fyn and/or ZAP-70 biology.

Response: We thank the reviewer for this suggestion. We agree that higher resolution images with live-cell analysis should advance our understanding of ZAP70 biology. We tried higher resolution imaging of the subcellular immunological synapse of TCR and CARs using a Spinning disk confocal in our core facility which is capable of FRET imaging. TCR engagement caused a ZAP70 activation, as visualized by our new FRET biosensor, with well-organized spatial distribution, concentrating at the central TCR cluster in the immunological synapse region and at the cell periphery where cortical actin fibers accumulate. This is significantly different from that of

CAR stimulation, which led to a relatively spread distribution of CAR clusters and ZAP70 activation patterns (New Supplementary Fig. 9g-h). We have added these new results into the revised manuscript (Results P10).

New Supplementary Figure. 9. Spatial distribution of activated ZAP70 in TCR- or CAR synapses.

g. Schematics of solid supported lipid bilayer modified with anti-CD3 and ICAM-1 (the cartoon on the left). The images on the right show the spatial distributions of Actin (labeled by LifeAct), TCR (by anti-CD3), ZAP70 localization (by biosensor intensity), and ZAP70 activity (by FRET ratio) in T cells, as indicated.

h. Schematics of solid supported lipid bilayer modified with CD19 and ICAM-1 (the cartoon on the left). The images on the right show the spatial distribution of Actin (labeled by LifeAct), CAR (labeled by an anti-mouse IgG, F(ab')₂), ZAP70 localization (by biosensor intensity), and ZAP70 activity (by FRET ratio) in T cells, as indicated.

3) The screen shown in Figure 5 found several inhibitors that reduced ZAP-70 kinase function. They then tested two of these compounds, staurosporine and AZD7762, in in vivo assays using Jurkat T cells. These compounds were able to reduce TCR-induced phosphorylation of ZAP-70 and LAT in cells expressing WT ZAP-70 and R360P ZAP-70. The authors then conclude that “the inhibitors identified through the saFRET-HTDS assay can efficiently inhibit the ZAP70 kinase signaling pathway” and state that these inhibitors “have therapeutic potentials in treating diseases involving abnormal ZAP70 kinase or T cell activation.” They also state in the abstract that “the improved saFRET biosensors further allowed the identification of compounds that demonstrated novel efficiency in inhibiting ZAP70 kinase activity and disease-related T-cell activation.” Staurosporine and AZD7762 certainly inhibit ZAP-70, but they also broadly inhibit many other kinases that are relevant for TCR signaling pathways. In fact, staurosporine is well known as a general tyrosine kinase inhibitor. The reduced phosphorylation of ZAP-70 and LAT by staurosporine and AZD7762 could be due to their inhibition of LCK, FYN and several other kinases.

Response: We thank the reviewer for this suggestion. We agree that the inhibitors that we identified may have off-targets. However, the FRET change induced by the identified inhibitors was specifically mediated by the change of the ZAP70 kinase domain. In fact, saFRET-HTDS enables the screening of inhibitors directly targeting ZAP70 Kinase with high specificity compared to conventional assays. We have performed additional experiments to verify that our saFRET-HTDS platform can

differentiate inhibitors of ZAP70 upstream molecules from those directly targeting ZAP70 itself, overcoming the drawback of the conventional FRET assays. We tested the inhibitors of signaling molecules acting upstream to ZAP70, e.g., Src, Fyn, and Lck kinases. Our results indicate that these non-specific inhibitors targeting kinases upstream to ZAP70 can be falsely selected by the conventional FRET assay (New Supplementary Fig. 10e-f), but not by the saFRET-HTDS assay (New Fig. 6c and New Supplementary Fig. 10g-h), demonstrating the high specificity of our saFRET-HTDS screening approach. We have added these new results and discussions into the revised manuscript (Results P11-12; discussion P14).

New Sup Figure 10. The saFRET assay is specific and immune to upstream signaling of the target kinase.

e, Schematics of conventional FRET assay in T cells. The FRET change could be affected by inhibitors targeting ZAP70 kinase and its upstream molecules.

f, The Src, Fyn, Lck, and Syk kinase inhibitors can significantly reduce the FRET change of conventional biosensors in response to CD3/CD28 antibody stimulation. The inhibitor groups were pretreated with corresponding inhibitors as indicated for 30 min before antibody stimulation. Control represents the DMSO pre-treated group. (n>12 in each group, One-way ANOVA, ****P<0.0001 and **P=0.0013).

g, Schematics of saFRET assay in HEK293 cells. The FRET change is mediated by the kinase domain.

h, Inhibitors of Src, Fyn, Lck, and Syk kinases could not cause a significant FRET change of the ZAP70 saFRET biosensor. Control represents the DMSO treated group. (n>15 in each group and data were tested by One-way ANOVA).

In addition, these drugs do not have therapeutic potential. Staurosporine is highly toxic to multiple cells and AZD7762 was withdrawn from clinical trials due to cardiotoxic effects. The authors show the toxic effects of these drugs in Figure 6F and 6L, where there is a large change in side scatter after treatment, and in Supplemental Video 7, where the drugs cause a marked change in cell morphology.

Response: We agree with the reviewer that staurosporine and AZD7762 are multi-target kinase inhibitors and have cell toxicity, limiting their therapeutic potentials. Staurosporine could also induce the cell morphology change. However, the FRET change is specific to the ZAP70 kinase change since the screening is done in a short time-window. To examine whether the inhibitor-induced FRET changes could be caused by cytotoxicity, we performed the live-cell imaging experiment to monitor cell cytotoxicity after inhibitor treatment (New supplementary Fig.11c-d). In this experiment, an AOPI dye was utilized to label the live (AO dye, Green) or dead cells (PI dye, Red). In the positive control group, significant cell death was observed as the cell nucleus was gradually labeled by PI dye after H₂O₂ treatment. However, we did not observe any cell death signal after 30 minutes of staurosporine addition with the dosage used in our experiments, when the FRET ratio was significantly reduced. These results suggest that our saFRET platform can identify inhibitors specifically for ZAP70, independent of cytotoxicity. We agree with the reviewer that staurosporine and AZD7762 may not have therapeutic potential at the current stage and thus moved some of the related data into supplementary figures to streamline the presentation. It is possible, however, that modifications of these inhibitors [e.g. conjugated to liposome nanoparticles¹⁷ or gold nanoparticles¹⁸] could enhance the specificity and reduce the cytotoxicity in the future.

To address the reviewer's critiques, we further examined Sunitinib (SUTENT), which is an FDA approved drug for renal cell carcinoma (RCC) and imatinib-resistant gastrointestinal stromal tumor (GIST) and ranked 4th in our saFRET screening assay (Fig. 6d). Similar to staurosporine and AZD7762, Sunitinib significantly inhibited the phosphorylation of ZAP70 and its downstream signal molecule LAT(Y191) (New Fig. 7 a-d for Jurkat, New Fig. 7 i-k for P116-ZAP70-R360P mutant and new Supplementary Fig. 12), as well as the subsequent activation of T cells represented by CD69 (New Fig. 7 e-f for Jurkat and New Fig. 7l for P116-ZAP70-R360P). Hence, sunitinib identified by our saFRET screening may be applicable to mitigate abnormal ZAP70 activation or ZAP70R360P related autoimmune diseases. Since sunitinib is an FDA approved drug whose safety has been demonstrated in human subjects, we expect that sunitinib has high therapeutic potential to be repurposed in treating ZAP70-related autoimmune diseases (e.g., R360P mutation has been shown to cause autoimmune disease requiring allogeneic hematopoietic cell transplantation in patients)¹. We have added the above results/discussions into the revised manuscript, including Figure 7 and supplementary figure 11-12.

While the selection of potent ZAP70 inhibitors, with high specificity, is extremely challenging, the screening of large FDA-approved drug libraries in the future may allow the identification of more high-performance inhibitors of ZAP70 kinase with therapeutic potentials. We expect that the counter-sorting strategy using our saFRET assay for the potential off-targets (e.g., Src, Lck, SYK kinase) should further allow the elimination of non-specific inhibitors. We have added the above discussions into the revised manuscript (P15).

New Supplementary Figure. 11. The inhibitor-induced FRET changes in our saFRET assay are not due to cytotoxicity.

c-d, Representative images (c) and time courses (d) of the green (live cell staining, AO dye) and red (dead cell staining, PI dye) channel signals of the cells under different treatments. H₂O₂ was used as a positive control to trigger cell death. No dead cell was observed during the time window of drug testing after the staurosporine treatment (10 μg/ml, same concentration to that in saFRET experiments). Error bars, mean ± SEM. Scale bars, 20 μm.

New Figure 7 (part1). Inhibition of T cell activation by Sunitinib.

a, Experimental scheme and timeline for experiments in b-d. Jurkat T cells were pre-treated with inhibitors for 30 min before anti-TCR stimulation by anti-CD3/CD28 antibodies for 5 min.

b, Immunostaining images of pLAT (Y191) in Jurkat T cells with sunitinib pre-treatment. Scale bars, 10 μ m.

c, Quantification of pLAT (Y191) intensity of single cells in different groups. ($n > 150$ for each group, One-way ANOVA, **** $P < 0.0001$). NC represents Jurkat T cells without any treatment, PC represents Jurkat T cells stimulated with anti-TCR only. Data was normalized to PC group. Error bars, mean \pm SD.

d, Quantification of pZAP70 (Y493) intensity of single cells in different groups. ($n > 200$ for each group, One-way ANOVA, **** $P < 0.0001$). Error bars, mean \pm SD.

i, Quantification of pZAP70 (Y493) intensity of single cells in different P116-ZAP70 R360P cells with different inhibitor pre-treatments. groups. ($n > 100$ for each group, One-way ANOVA, **** $P < 0.0001$). Error bars, mean \pm SD.

j, Images of pLAT (Y191) in P116-ZAP70 R360P cells with sunitinib pre-treatment. Scale bars, 10 μ m.

k, Quantification of pLAT (Y191) intensity of P116-ZAP70 R360P cells with different inhibitor pre-treatments. ($n > 150$ for each group, One-way ANOVA, **** $P < 0.0001$). Error bars, mean \pm SD.

New Supplementary Fig. 12. Sunitinib is a potent inhibitor of ZAP70 signaling pathway.

a, Representative images of pZAP70 (Y493) in stimulated Jurkat T cells with sunitinib pretreatment. Scale bars, 10 μ m.

b, Representative images of pZAP70 (Y493) in stimulated P116-ZAP70-R360P cells with sunitinib pretreatment. Scale bars, 10 μ m.

New Figure 7 (part2). Inhibition of T cell activation by Sunitinib.

e, Experimental scheme and timeline for CD69 staining experiment.

f, Flow-cytometry analysis of CD69 expression in T cells after anti-TCR stimulation, with or without sunitinib pre-treatment.

l, Flow-cytometry analysis of CD69 expression in P116-ZAP70-R360P cells after anti-TCR stimulation, with or without sunitinib pre-treatment. ZAP70-R360P expression levels in P116 cells were indicated by YPet intensity.

In the end, the screen is novel and can identify inhibitors of ZAP-70 but the inhibitors identified in the described screen also inhibit many other kinases. Figure 6 has little relevance other than to confirm that staurosporine and AZD7762 are good kinase inhibitors and the statements describing this work are not supported by the data.

Response: We agree with the reviewer that staurosporine and AZD7762 may have off-targets in T cells, and have changed the statement accordingly. In Figure 6 (now figure 7), we aim to show that the inhibitors identified from the saFRET assay can be used in inhibiting the abnormal T cell activation originated from hyper ZAP70 activities. As we have shown above (e.g., the new Supp Fig. 10), the inhibitors identified by our saFRET assay should directly inhibit ZAP70 kinase activities when tested in T cells. The selection of staurosporine and AZD7762, despite their potential cytotoxicity in long terms, demonstrated the successful application of the saFRET-HTDS platform in identifying ZAP70 inhibitors. We also discovered that an FDA-approved drug Sunitinib, which was originally approved to treat renal cell carcinoma (RCC) and imatinib-resistant gastrointestinal stromal tumor (GIST), can be repurposed to inhibit the abnormal ZAP70 kinase or its disease-related hyperactive ZAP70 R360P mutants (New Fig.7). Since sunitinib did not affect the activation of Lck, the direct upstream effector of ZAP70¹⁹, sunitinib can be a relatively specific ZAP70 kinase inhibitor in T cells with future therapeutic potentials. Nevertheless, we agree with the reviewer that inhibitors selected from our saFRET assay may still need counter screenings against other kinases in the future to ensure their specificity. Accordingly, we have added the above discussions in the revised manuscript (Page 15).

Reference:

1. Chan, A.Y. et al. A novel human autoimmune syndrome caused by combined hypomorphic and activating mutations in ZAP-70. *J Exp Med* **213**, 155-165 (2016).
2. Yan, Q. et al. Structural basis for activation of ZAP-70 by phosphorylation of the SH2-kinase linker. *Mol Cell Biol* **33**, 2188-2201 (2013).
3. Randriamampita, C. et al. A novel ZAP-70 dependent FRET based biosensor reveals kinase activity at both the immunological synapse and the antisynapse. *PLoS one* **3**, e1521-e1521 (2008).
4. Cadra, S. et al. ROZA-XL, an improved FRET based biosensor with an increased dynamic range for visualizing zeta associated protein 70 kD (ZAP-70) tyrosine kinase activity in live T cells. *Biochem Biophys Res Commun* **459**, 405-410 (2015).
5. Sadelain, M. CD19 CAR T Cells. *Cell* **171**, 1471 (2017).
6. Feucht, J. et al. Calibration of CAR activation potential directs alternative T cell fates and therapeutic potency. *Nat Med* **25**, 82-88 (2019).
7. Rohrs, J.A., Siegler, E.L., Wang, P. & Finley, S.D. ERK Activation in CAR T Cells Is Amplified by CD28-Mediated Increase in CD3 ζ Phosphorylation. *iScience* **23**, 101023 (2020).
8. Gudipati, V. et al. Inefficient CAR-proximal signaling blunts antigen sensitivity. *Nat Immunol* **21**, 848-856 (2020).
9. Komatsu, N. et al. Development of an optimized backbone of FRET biosensors for kinases and GTPases. *Molecular biology of the cell* **22**, 4647-4656 (2011).
10. Ellis, E.L. & Delbrück, M. THE GROWTH OF BACTERIOPHAGE. *The Journal of general physiology* **22**, 365-384 (1939).
11. Wang, P. et al. Visualizing Spatiotemporal Dynamics of Intercellular Mechanotransmission upon Wounding. *ACS photonics* **5**, 3565-3574 (2018).
12. Zhang, J.-F. et al. An ultrasensitive biosensor for high-resolution kinase activity imaging in awake mice. *Nature Chemical Biology* **17**, 39-46 (2021).
13. Songyang, Z. et al. SH2 domains recognize specific phosphopeptide sequences. *Cell* **72**, 767-778 (1993).
14. Songyang, Z. et al. Catalytic specificity of protein-tyrosine kinases is critical for selective signalling. *Nature* **373**, 536-539 (1995).
15. Xiang, X. et al. A FRET-Based Biosensor for Imaging SYK Activities in Living Cells. *Cell Mol Bioeng* **4**, 670-677 (2011).
16. Li, K. et al. Imaging Spatiotemporal Activities of ZAP-70 in Live T Cells Using a FRET-Based Biosensor. *Annals of biomedical engineering* **44**, 3510-3521 (2016).
17. Mukthavaram, R. et al. High-efficiency liposomal encapsulation of a tyrosine

kinase inhibitor leads to improved in vivo toxicity and tumor response profile. *Int J Nanomedicine* **8**, 3991-4006 (2013).

18. Gossai, N.P. et al. Drug conjugated nanoparticles activated by cancer cell specific mRNA. *Oncotarget* **7** (2016).
19. Gu, Y. et al. Sunitinib impairs the proliferation and function of human peripheral T cell and prevents T-cell-mediated immune response in mice. *Clinical Immunology* **135**, 55-62 (2010).

Reviewers' Comments:

Reviewer #1:

Remarks to the Author:

Accept for publication.

Reviewer #2:

Remarks to the Author:

Liu and colleagues present a revised manuscript on engineering of FRET sensors via mammalian cell based libraries and FACS. My major concerns were not sufficiently addressed. Thus, I am not convinced that we see a general and systematic method to engineer high performing FRET sensors. Major key strategic sites for engineering in those sensors, e.g. linker sequences, were not touched. Similarly, any potential interfaces between fluorescent proteins that might modulate FRET via changes in stickiness were not considered. Thus, the claim to present a systematic approach appears to be broad and overstated. The sensors that were identified are improved compared to parental versions but apparently have smaller FRET changes than other state-of-the-art FRET sensors.

Nevertheless, the manuscript has lots of merit: The sensors for Fyn and ZAP70 are better than previous versions, the principle of using self-activating sensors may be useful for engineering similar FRET sensors, and the characterization of the biosensors provides interesting insights. Thus, if the manuscript could be narrowed down in scope and claims, with a revised title and abstract reflecting this, it could be acceptable.

Reviewer #3:

Remarks to the Author:

I thank the authors for a careful and thoughtful response to my comments. The authors have addressed all my major comments with new experiments that have added greatly to the impact of their method to the field.

Response to the reviewers

We sincerely thank the editor and reviewers for the effort and help in evaluating our manuscript as well as the positive affirmation and the suggestions for the improvement of our manuscript.

REVIEWER COMMENTS

Reviewer #1 (Remarks to the Author):

Accept for publication.

Response: We appreciate the reviewer's effort in evaluating our manuscript, and we also thank the reviewer for the affirmation of our revised manuscript.

Reviewer #2 (Remarks to the Author):

Liu and colleagues present a revised manuscript on engineering of FRET sensors via mammalian cell based libraries and FACS. My major concerns were not sufficiently addressed. Thus, I am not convinced that we see a general and systematic method to engineer high performing FRET sensors. Major key strategic sites for engineering in those sensors, e.g. linker sequences, were not touched. Similarly, any potential interfaces between fluorescent proteins that might modulate FRET via changes in stickiness were not considered. Thus, the claim to present a systematic approach appears to be broad and overstated. The sensors that were identified are improved compared to parental versions but apparently have smaller FRET changes than other state-of-the-art FRET sensors. Nevertheless, the manuscript has lots of merit: The sensors for Fyn and ZAP70 are better than previous versions, the principle of using self-activating sensors may be useful for engineering similar FRET sensors, and the characterization of the biosensors provides interesting insights. Thus, if the manuscript could be narrowed down in scope and claims, with a revised title and abstract reflecting this, it could be acceptable.

Response: We appreciate the reviewer's effort in evaluating our revised manuscript, and providing helpful suggestions for the further improvement of our manuscript.

We agree with the reviewer that more work can be further conducted using our platform to optimize linkers, fluorescent proteins, and other components (e.g., binding domains) before a general and systematic method can be fully established to engineer and develop biosensors, although our current work focusing on substrate sequence is also crucial for improving biosensor

performance. Therefore, we followed the reviewer's suggestions to narrow down the scope and claims by removing the wordings such as "general" and "systematic" after carefully reviewing the whole manuscript. Specifically, we have changed the title from "FRET-Seq: Integration of FRET and Sequencing to Engineer Biosensors from Mammalian Cell Libraries" to "FRET-Seq: Integration of FRET and Sequencing to Engineer *Kinase* Biosensors from Mammalian Cell Libraries", to make it more specific. Also, we changed "a systematic approach" into "an approach", "high-performance" into "enhanced performance", and specifically emphasized on the "substrate sequence" in the abstract. At the same time, we also changed the wording in the main text and discussion accordingly.

We have also added "While the focus of this work is on the substrate sequence of the biosensor, optimizations of linkers, SH2 domains, and fluorescent proteins may further enhance the performance of the biosensor and our FRET-Seq platform can be extended to screen more diverse libraries to optimize these important components of the FRET biosensors in the future." into the discussion part in the revised manuscript to reflect the change in narrowing down the scope/claims.

All the changes were highlighted in the revised manuscript.

Reviewer #3(Remarks to the Author):

I thank the authors for a careful and thoughtful response to my comments. The authors have addressed all my major comments with new experiments that have added greatly to the impact of their method to the field.

Response: We appreciate the reviewer's effort in evaluating our manuscript, and we also thank the reviewer for the affirmation of our revised manuscript.